# A new aerial approach for quantifying and attributing methane emissions: implementation and validation

**Jonathan F. Dooley**[1], **Kenneth Minschwaner**[1], **Manvendra K. Dubey**[2], **Sahar H. El Abbadi**[3,a], **Evan D. Sherwin**[3,a], **Aaron G. Meyer**[2,b], **Emily Follansbee**[2], **and James E. Lee**[2]

[1]Department of Physics, New Mexico Institute of Mining and Technology, Socorro, NM 87801, USA
[2]Earth and Environmental Sciences, Los Alamos National Laboratory, Los Alamos, NM 87545, USA
[3]Department of Energy Science & Engineering, Stanford University, Stanford, CA 94305, USA
[a]present address: Sustainable Energy Department, Lawrence Berkeley National Laboratory, Berkeley, CA 94720, USA
[b]present address: Energy & Geoscience Institute, The University of Utah, Salt Lake City, UT 84112, USA

**Correspondence:** Jonathan F. Dooley (jonathan.dooley@student.nmt.edu)

**Abstract.** Methane ($CH_4$) is a powerful greenhouse gas that is produced by a diverse set of natural and anthropogenic emission sources. Biogenic methane sources generally involve anaerobic decay processes such as those occurring in wetlands, melting permafrost, or the digestion of organic matter in the guts of ruminant animals. Thermogenic $CH_4$ sources originate from the breakdown of organic material at high temperatures and pressure within the Earth's crust, a process which also produces more complex trace hydrocarbons such as ethane ($C_2H_6$). Here, we present the development and deployment of an uncrewed aerial system (UAS) that employs a fast (1 Hz) and sensitive ($1$–$0.5\,\mathrm{ppb\,s^{-1}}$) $CH_4$ and $C_2H_6$ sensor and ultrasonic anemometer. The UAS platform is a vertical-takeoff, hexarotor drone (DJI Matrice 600 Pro, M600P) capable of vertical profiling to 120 m altitude and plume sampling across scales up to 1 km. Simultaneous measurements of $CH_4$ and $C_2H_6$ concentrations, vector winds, and positional data allow for source classification (biogenic versus thermogenic), differentiation, and emission rates without the need for modeling or a priori assumptions about winds, vertical mixing, or other environmental conditions. The system has been used for direct quantification of methane point sources, such as orphan wells, and distributed emitters, such as landfills and wastewater treatment facilities. With detectable source rates as low as 0.04 and up to $\sim 1500\,\mathrm{kg\,h^{-1}}$, this UAS offers a direct and repeatable method of horizontal and vertical profiling of emission plumes at scales that are complementary to regional aerial surveys and localized ground-based monitoring.

## 1 Introduction

Methane ($CH_4$) is the second-largest contributor to anthropogenic radiative forcing, with a global warming potential (GWP-20) greater than $80 \times$ that of carbon dioxide ($CO_2$) (Solomon et al., 2007; Schneising et al., 2020; Holmes et al., 2013). Methane is produced from many natural and anthropogenic sources which can be further classified as biogenic or thermogenic in origin. The largest biogenic sources result from anaerobic decay such as wetlands, melting permafrost, or the breakdown of organic matter in the guts of ruminant animals. Thermogenic $CH_4$, "natural gas", is generated during the breakdown of organic matter at high temperatures and pressure within the Earth's crust. The later process also produces more complex trace hydrocarbons such as ethane and propane, collectively known as natural gas liquids (NGLs). Coal mining, natural seeps, oil and natural gas (O&NG) activities, and supply chain leakage each produce varying amounts of NGLs. While sometimes other gases predominate (e.g., $CO_2$), all thermogenic sources contain some fraction ethane ($C_2H_6$) (Kutcherov and Krayushkin, 2010; Glasby, 2006; Etiope and Sherwood Lollar, 2013), and it is the second-most prevalent NGL in natural gas processed for energy generation (Hodnebrog et al., 2018; Solomon et al.,

2007; Simpson et al., 2012; Peischl et al., 2018; Karion et al., 2015; Johnson et al., 2019). Biogenic methane emissions do not contain ethane (Masson-Delmotte et al., 2023; Cheewaphongphan et al., 2019), so $C_2H_6$ is therefore a critical marker for source attribution, particularly when distinguishing between biogenic and thermogenic methane emissions (Solomon et al., 2007; Karion et al., 2015; Johnson et al., 2019). Accurately quantifying contributions from various sources, both spatially and temporally, is an important step towards building local, regional, and global $CH_4$ emission estimates as well as informing policy decisions.

The accuracy of regional and global methane emission estimates is limited due to the sheer number of unknown variables. There are two primary methods to estimate methane emissions: statistical analysis of source activity inventories, known as "bottom-up" (BU) estimations, and "top-down" (TD) estimates based on observations from instruments deployed on satellite, aircraft, and ground-based platforms (Heath et al., 2015; Schneising et al., 2020; Vaughn et al., 2018; Cheewaphongphan et al., 2019; Frankenberg et al., 2016). A variety of BU and TD methods have been used to detect methane leaks (Frankenberg et al., 2005) and inform policy decisions (Heath et al., 2015), but persistent discrepancies remain between the BU and TD methodologies (Masson-Delmotte et al., 2023; Solomon et al., 2007; Vaughn et al., 2018; Peischl et al., 2018; Cheewaphongphan et al., 2019). In general, BU and TD estimation methods only represent a snapshot in time and do not effectively account for temporal variations due to diurnal, seasonal, and activity cycles. Recent studies have shown that O&NG methane emissions vary significantly in time (Lavoie et al., 2015; Johnson et al., 2019; Vaughn et al., 2018), showing that more frequent and repeated measurements are important in helping to reduce uncertainties in emission monitoring (Frankenberg et al., 2005; Space Studies Board et al., 2019).

High-altitude and space-based systems offer robust methods of monitoring $CH_4$ and other greenhouse gases (GHGs) at regional and global scales (Sherwin et al., 2024; Schneising et al., 2020). However, most large-area satellites (Sentinel, Landsat) drop off in detection at $\sim 1000\,\text{kg}\,\text{h}^{-1}$ with only the very sophisticated or targeted systems able to quantify sources $< 500\,\text{kg}\,\text{h}^{-1}$ (Sherwin et al., 2024). These instruments for TD emission estimates are ideally supplemented with contemporaneous ground-based measurements to constrain the monitoring capabilities – e.g., Total Carbon Column Observing Network, TCCON (Vaughn et al., 2018; Pétron et al., 2020; Kort et al., 2014; Parker et al., 2011; Turner et al., 2015; Gisi et al., 2012).

A common method for estimating GHG emissions is a mass balance approach, which uses GHG concentration measurements upwind and downwind of target sources to isolate the source from background concentrations (Frankenberg et al., 2016; Schwietzke et al., 2017; Johnson et al., 2019). These measurements can be collected from aircraft instruments along upwind and downwind transects (Franken-

berg et al., 2016; Schwietzke et al., 2017) or by simultaneous measurements using similar ground-based instruments installed upwind and downwind of the sources (Gisi et al., 2012; Saad et al., 2014; Heerah et al., 2021). The accuracy of the latter method depends on relatively constant wind speed and direction during data collection. Therefore, accurate knowledge of principal wind conditions is required for proper instrument installation, and only those data obtained under favorable conditions can be used to accurately estimate the flux (Gisi et al., 2012; Lavoie et al., 2017). Aircraft quantification methods generally drop off around $10\,\text{kg}\,\text{h}^{-1}$ for the highly sensitive instruments, so it is difficult or impossible to quantify low-emitting sources using combined ground, aircraft, and satellite data to constrain and validate the estimates. Ground-based detection systems are the most straightforward and accessible methods, but effective site monitoring is highly dependent on wind directions. Column-averaged measurement techniques using Fourier transform spectrometers (FTSs) also require boundary layer height estimations and are unable to provide vertical profiles of source plumes.

There is a need for direct, repeatable, and cost-effective methods for detecting and quantifying $CH_4$ emissions from relatively small sources ($< 1\,\text{kg}\,\text{h}^{-1}$), which do not require a priori assumptions. Small uncrewed aerial vehicles (UAVs) offer new approaches to airborne air pollution and emission monitoring over scales and locations which are difficult to detect or access with other regional monitoring systems (Chen et al., 2024; Villa et al., 2016; Burgués and Marco, 2020). While designs can vary dramatically between models, UAVs are either fixed- or rotary-wing platforms. Fixed-wing UAVs are typically able to cover larger areas and generally allow more options for sensor mounting configurations, but they are unable to hover and quickly adapt to environmental conditions while tracking emission plumes. Rotary wing platforms – also known as vertical takeoff and landing (VTOL) or multirotor UAVs – generally have lower required operating velocities and have the ability to hover and can therefore be used for more complex, discontinuous missions at higher spatial resolution (McKinney et al., 2019; Villa et al., 2016; Burgués and Marco, 2020). Multirotor UAVs usually have between four and eight individual propellers, which to generate thrust and the power consumption results in shorter flight times than their fixed-wing counterparts, but they do not require specialized equipment or runways for takeoffs and landings. Recent advances in control technology have made multirotor systems easier to reliably operate.

A major issue with fixed-wing systems is that the higher operating velocities and minimum height requirements result in low detection probabilities during site surveys (Barchyn et al., 2019). Rotary UAV systems do not require a minimum velocity to stay aloft and can therefore be outfitted with a wider range of equipment for physical and chemical sensing (Hollenbeck et al., 2018; Shah et al., 2019; McKinney et al., 2019; Villa et al., 2016; Burgués and Marco, 2020). For instance, onboard anemometers for in situ wind

speed and direction have been shown to be more accurate onboard VTOL platforms, while higher relative winds and aerodynamic flows around fixed-wing platforms often result in less accurate in situ measurements (Hollenbeck et al., 2018). While multirotor UAVs have a significant propeller wash effect below the body, induced winds are negligible when anemometers are mounted above the shallow inflow layer (Barbieri et al., 2019; Hollenbeck et al., 2018; Villa et al., 2016; Barchyn et al., 2019).

The fast development of commercial UAVs and low-weight sensors has driven a multitude of scientific studies which would have been difficult or impossible to conduct over a decade ago. Both inverse modeling and mass-balance approaches have been used to calculate total emissions across a wide range of spatial scales. McKinney et al. (2019) deployed a hexarotor UAV outfitted with adsorbent cartridges to collect biogenic volatile organic compound (VOC) emissions at various locations in central Amazonia. Shah et al. (2019) calculated 3D flux densities by fitting direct measurements of downwind $CH_4$ plumes to near-field Gaussian plume models. Olaguer et al. (2022) used contemporaneous UAV and mobile ground-based measurements to estimate biogenic $CH_4$ emissions from a landfill. Bel Hadj Ali et al. (2020) compared multiple ground-based emission monitoring techniques with downwind plume measurements fitted to Gaussian plume models. Gålfalk et al. (2021) piloted a small quadcopter to fully surround a known biogenic methane hotspot for mass balance emission estimates $< 200\,kg\,h^{-1}$.

This study describes the development and implementation of a new uncrewed aerial system (UAS) to address critical areas of research by (a) accurately and directly quantifying emissions on small spatial and temporal scales, especially in hard to reach places; (b) constraining BU inventories driven by small-scale estimations; and (c) separating and attributing thermogenic and biogenic sources. The New Mexico Tech (NMT) UAS design combines rapid sampling of chemical and meteorological data with a mobile platform capable of vertical and horizontal profiling relative to target sources for direct and high-resolution sampling of emission plumes. A commercially available UAV is outfitted with a multi-sensor onboard payload including a compact mid-IR spectrum analyzer and lightweight anemometer for efficient, repeatable quantification and characterization of various localized anthropogenic sources.

## 2 Methods

### 2.1 System design

The full uncrewed aerial system (UAS) is depicted in Fig. 1. It includes four main components: (1) a mobile mid-IR methane and ethane sensor (Aeris MIRA Pico), (2) a lightweight 3D vector wind and environmental sensor (Anemoment TriSonica Mini), (3) an uncrewed aerial vehicle (DJI Matrice 600 Pro, M600P), and (4) an onboard computer for system monitoring and data collection.

#### 2.1.1 Methane and ethane sensor

The MIRA (Mid-InfraRed Analyzer) Pico leak detection system (LDS) developed by Aeris Technologies employs a solid-state laser and multi-pass absorption cell with a spectral band pass between 2.5 and 4.7 µm. The wide spectral range allows for simultaneous measurements of both $CH_4$ and $C_2H_6$ mixing ratios with a precision of $1\,ppb\,s^{-1}$ and $0.5\,ppb\,s^{-1}$, respectively (Aeris Technologies Inc, 2019; Scherer, 2017). The MIRA Pico is outfitted with a flexible length of tubing (see Sect. 2.1.4), and the constant pump flow rate results in a phase lag of $\sim 2\,s$ based on laboratory tests of the response delay from pulsed gas releases (Sect. 2.1.4). The MIRA has been used extensively on the ground, including controlled-release tests at Colorado State University's (CSU) METEC facility to find and quantify leaks (Travis et al., 2020).

#### 2.1.2 Onboard anemometer and weather sensor

In addition to hydrocarbon concentrations, direct flux quantification requires measurements of vector winds, temperature, and pressure. The system includes an ultra-light 3D sonic anemometer: the TriSonica Mini weather sensor (TWS) by Anemoment has a mass of 50 g with a volume of less than 450 cm$^3$ (Anemoment LLC, 2021). Additionally, environmental measurements can also be used as a way to detect and adapt to unsafe conditions during flights (Hollenbeck et al., 2018). The TWS senses vector winds ($|\boldsymbol{u}|$), temperature ($T$), and pressure ($P$) at 5 Hz with an accuracy of $\delta u_m = \delta v_m = 0.2\,m\,s^{-1}$, $\delta T = 2\,C$, and $\delta P = 10\,hPa$, respectively (see Table 1).

#### 2.1.3 Uncrewed aerial vehicle

The DJI Matrice 600 Pro (M600P) is a hexarotor vertical takeoff and landing aircraft that is capable of flying with relatively large payload masses of up to 5.5 kg. This powerful and mobile platform became commercially available in 2014 and has been used in a variety of scientific and commercial applications (McKinney et al., 2019; Villa et al., 2016; Hollenbeck et al., 2018). The M600P can be remotely piloted at a distance of up to 5 km at a maximum altitude of 125 m above ground level. Flight times depend on factors such as payload mass, winds, and flight pattern. Under typical winds of 2–4 m s$^{-1}$ and the standard total payload mass of $\sim 3\,kg$, total flight times range between 18 and 25 min. We employ dual battery packs that are cycled between flying and charging in the field; due to battery life and charging limitations, the UAS typically can accommodate about one flight per hour. Additionally, the flux quantification method described in Sect. 2.6 relies on relatively stable wind fields around the source, optimally between 2 and 6 m s$^{-1}$.

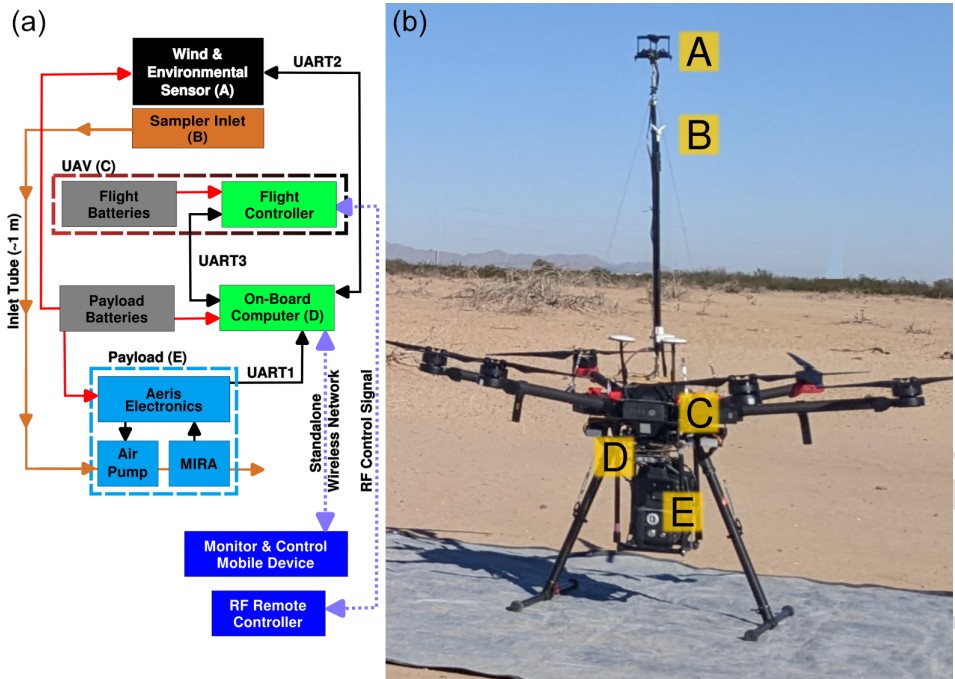

**Figure 1.** Uncrewed aerial system (UAS) design overview. **(a)** System block diagram: (A) TriSonica Mini weather sensor (TWS), (B) gas sampler inlet, (C) Matrice 600 Pro uncrewed aerial vehicle (M600P), (D) Raspberry Pi 4 onboard computer, and (E) Aeris MIRA Pico methane and ethane sensor. **(b)** Flight-ready UAS collecting ground data prior to takeoff. TWS and sampler inlet (A and B) mounted on a carbon fiber tube mast above the M600P's propeller inflow layer ("rotor wash").

**Table 1.** Sensor uncertainties for error propagation (Anemoment LLC, 2021; Dà Jiāng Innovations, 2018; Aeris Technologies Inc, 2019).

| Sensor | Measurement | Uncertainty | Units | Variable |
|---|---|---|---|---|
| TWS | Pressure | 1000 | Pa | $\delta P$ |
| TWS | Temperature | 2 | C | $\delta T$ |
| TWS | Wind speed | 0.2 | $\mathrm{m\,s^{-1}}$ | $\delta u_{\mathrm{m}}, \delta v_{\mathrm{m}}$ |
| M600P GNSS | Horizontal velocity | 0.05 | $\mathrm{m\,s^{-1}}$ | $\delta V_x, \delta V_y$ |
| M600P IMU | Heading (yaw) | 0.05 | rad | $\delta\phi$ |
| M600P IMU | Pitch, roll | 0.017 | rad | $\delta\theta, \delta\psi$ |
| MIRA Pico | Methane | 1.0 | ppb | $\delta\chi_{\mathrm{CH_4}}$ |
| MIRA Pico | Ethane | 0.5 | ppb | $\delta\chi_{\mathrm{C_2H_6}}$ |
| RPi RTC | Timestamp | 100 | ms | $\delta(\Delta t)$ |
| Pipeline* | Methane background | 5.0 | ppb | $\delta\chi_{0,\mathrm{CH_4}}$ |
| Pipeline* | Ethane background | 1.0 | ppb | $\delta\chi_{0,\mathrm{C_2H_6}}$ |

\* Average $1\sigma$ standard deviation of residuals from background estimation ($\varepsilon$); actual uncertainty is dependent on the quality of the background estimate ($\chi_0$) for each of the individual datasets.

The M600P is controlled via proprietary software and firmware with an expansive application programming interface (API) for telemetry logging and flight control (Dà Jiāng Innovations, 2018). However, the reliance on proprietary software limits system customizability and sensor integration (see Sect. 2.1.5). While it is possible to automate flight plans for the M600P, it is more important for the operator to maintain control of the UAS throughout the entire flight to account for changing or unexpected flight conditions.

### 2.1.4 Payload sensor mounting

Both the Raspberry Pi (RPi) onboard computer and the MIRA Pico mount to the underside of the M600P as shown in Fig. 1. The M600P achieves flight by funneling air downwards to create thrust, constantly displacing air around the vehicle's body in the process (Dà Jiāng Innovations, 2018). Computational fluid dynamics simulations carried out by McKinney et al. (2019) determined that the Matrice 600 Pro causes disturbances up to 5 m below the M600P's center of

mass; however, the magnitude of the disturbances drops off quickly at a height of 60 cm above the M600P body (McKinney et al., 2019).

The TWS mounts to a mast made of lightweight, carbon fiber tubing (0.5 in diameter) anchored near the M600P's center of mass. This places the TWS $\sim 85$ cm above the M600P and above the shallow inflow layer generated by propellers during normal flight. At this height above the propellers, air disturbances are below the noise floor of the TWS ($\sim 0.2\,\mathrm{m\,s^{-1}}$) or easily calibrated out during data processing. A $\sim 1$ m section of flexible tubing connects the payload's gas cell to a sampling port approximately 80 cm above the body of the M600P. This physical offset between the MIRA and sampling port introduces a phase lag of $\sim 2$ s between the measured MIRA concentrations and the TWS, which is accounted for prior to the calculation of instantaneous fluxes. The sampler port is outfitted with a plastic Y-shaped inlet for redundant openings to decrease the chance of damaging the pump due to debris clogging the inlet.

While the TWS has an integrated magnetometer to measure heading, this device is intended for stationary deployment in an open location where averaging of compass readings can result in a relatively accurate direction. However, any magnetic disturbances created from adjacent devices or operating motors perturb the heading measurements. In contrast, the M600P has a triple-redundant positioning system with an accuracy of $< 3°$ at 200 Hz. The TWS must be physically aligned with the more accurate heading data from the M600P's inertial measurement unit (IMU) to accurately calculate static-frame vector winds (more in Sect. 2.2).

In order to ensure proper alignment, the M600P is set up facing north with the TWS mounting point tightened so that the anemometer's north arm is facing the same direction. The MIRA Pico and onboard computer are both mounted under the body of the M600P as shown in Fig. 1. Initial M600P test flights revealed that payload mounting at this location improves flight performance, especially during takeoff and landing, due to the lower center of mass.

### 2.1.5 Onboard data logging and transmission

Each of the three main devices (MIRA Pico, TWS, and M600P) has its own output data stream that must be separately parsed and then temporally synchronized with the other streams. The M600P does not support the use of DJI's proprietary "Payload SDK" – firmware enabling sensor data to be transmitted along with the radio frequency control signal. Therefore, remote monitoring and control of the fully integrated UAS are handled independently of the M600P via an onboard device for both acquisition and transmission. A fourth-generation Raspberry Pi (RPi) single-board computer is the ideal choice due to its size, weight, and programmability via the Linux Kernel. The RPi is powered independently via a 10 Ah payload battery (see Fig. 1), allowing for ground-based data collection between flights, typically while the M600P batteries are charging.

## 2.2 Relative wind adjustments

The TriSonica Mini weather sensor (TWS) is designed for static installation with one arm facing north (Anemoment LLC, 2021) and will therefore produce incorrect results when the device's heading ("yaw", $\phi$) is not a multiple of $2\pi$ (Anemoment LLC, 2021; Hollenbeck et al., 2018). In addition, the motion of the M600P during data collection induces an apparent wind that is folded into the TWS wind measurement. Transforming the raw TWS vector wind measurements ($\boldsymbol{u}_\mathrm{m} = [u_\mathrm{m}, v_\mathrm{m}, w_\mathrm{m}]$) to Earth-fixed coordinates therefore requires accurate, real-time measurements of $\phi$ and UAS velocity ($\boldsymbol{V}_\mathrm{s}$). In order to transform the measured TWS data to a static, Earth-fixed coordinate system, we apply a standard Galilean transformation as shown in Eq. (1).

$$
\begin{aligned}
\boldsymbol{u} &= \mathbf{R}(\phi)\boldsymbol{u}_\mathrm{m} + \boldsymbol{V}_\mathrm{s} \\
&= \begin{bmatrix} \cos(\phi) & \sin(\phi) & 0 \\ -\sin(\phi) & \cos(\phi) & 0 \\ 0 & 0 & 1 \end{bmatrix} \begin{bmatrix} u_\mathrm{m} \\ v_\mathrm{m} \\ w_\mathrm{m} \end{bmatrix} + \begin{bmatrix} V_x \\ V_y \\ V_z \end{bmatrix} \\
&= \begin{bmatrix} u \\ v \\ w \end{bmatrix}
\end{aligned} \tag{1}
$$

Here, $\boldsymbol{u}_\mathrm{m} = [u_\mathrm{m}, v_\mathrm{m}, w_\mathrm{m}]$ are the TWS-measured vector winds, $\boldsymbol{u} = [u, v, w]$ is the corrected wind speed in Earth-fixed coordinates, and $\boldsymbol{V}_\mathrm{s} = [V_x, V_y, V_z]$ is the instantaneous UAS velocity. $\mathbf{R}(\phi)$ is the counterclockwise (CCW) Euler rotation matrix around the vertical axis ($\hat{z}$), where $\phi$ is the heading (yaw) of the M600P.

Rotations around the $\hat{y}$ and $\hat{x}$ axes – caused by variations in M600P pitch ($\theta$) and roll ($\varphi$), respectively – are both generally less than $3°$ during steady, level flights through the source plume and are therefore neglected in Eq. (1). The instantaneous pitch and roll angle can be significant when the M600P is being maneuvered to different altitudes between transects, but measurements collected during these spurious adjustments are filtered out during the flux quantification process (see Sect. 2.6).

A time series of the TWS-measured raw winds during a typical flight is shown in Fig. 2, along with the derived static-frame vector winds based on the M600P headings and velocities throughout the flight. The raw-wind measurements show clear signatures of back-and-forth, quasi-horizontal motions of the M600P during sampling of a methane plume. Winds adjusted for changes in heading and horizontal translation show a more realistic, continuous structure, but some residual effects can be discerned at turning points in the flight, when the platform's horizontal acceleration is large and the corresponding pitch and roll angles become appreciable. In general, pitch and roll angles during level, steady flight sections are less than $3°$ (see Fig. A1), but these angles can in-

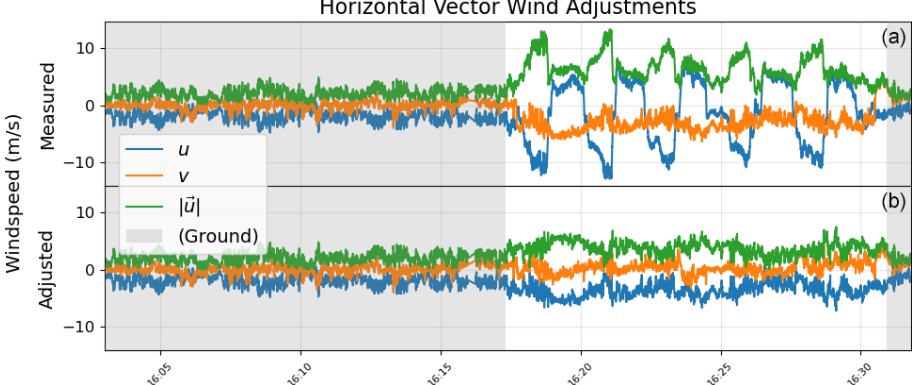

**Figure 2.** Raw measured vector winds (**a**) and heading- and velocity-corrected vector winds (**b**) before, during, and after flight. Pre- and post-flight ground wind speeds at the takeoff location highlighted in gray.

crease to 10° or more during rapid horizontal accelerations. We estimate that a 15° pitch or roll angle may introduce an error of up to 4 % in measured horizontal winds; therefore data obtained during turning points and deliberate changes
in flight altitude are screened from the analysis of methane fluxes (see Sect. 2.6). The adjusted and filtered winds have been compared to tower anemometer measurements during controlled-release validation experiments (Sect. 3), and these show agreement to within ± 15 % over ∼ 1 min averages and
with spatial separations of about 100 m. Individual wind vector components have also been compared in the field (see Fig. A2) when the UAS was flown between 10 and 30 m downwind of a static, single-component wind gauge. Comparisons of 5 s averages show agreement to within 10 % over
a range of winds between 2 and 7 m s$^{-1}$.

## 2.3  Background concentration estimation

While the MIRA Pico is highly sensitive with a large dynamic range (∼ 20 ppb < $\chi_{CH_4}$ < 4000 ppm) (Aeris Technologies Inc, 2019; Meyer et al., 2022; Follansbee et al.,
2024), there are observable levels of semi-periodic drift in the measured mole fractions for both hydrocarbons at the parts per billion level. This effect is especially noticeable for $\chi_{C_2H_6}$, partly because ambient $C_2H_6$ concentrations are generally ∼ 100 × smaller than $CH_4$. In addition to these instru-
mental drifts, the raw concentration measurements ($\chi$) may reflect actual changes in ambient background. Nearby point sources do not directly influence these background variations; instead they may be related to regional-scale emissions and meteorological influences such as winds and stability at
the mesoscale level. Both of these effects contribute to structure in the measured background ($\chi_0$), which must be accurately quantified in order to isolate enhancements from the target source. This background variability generally occurs over periods of minutes and is independent between $\chi_{CH_4}$
and $\chi_{C_2H_6}$, so direct comparison prior to background removal can be misleading, especially over longer period datasets.

The UAS measurement strategy involves repeated crosswind transects through the dispersing plume downwind of a target source (as discussed in more detail in Sect. 2.5). The resulting time series concentration data contain a num-
40 ber of finite-width enhancements superimposed on a slowly varying background, as shown in Fig. 3a. Over the course of these measurements, the solar insolation was increasing and the wind speed and direction were changing, which led to a monotonic increase in the background methane concen-
45 tration as seen with the increasing background. It should be noted that each plume "spike" in Fig. 3a should not be expected to have the same peak height or width, since these plume intersects all occur at different times, altitudes, and downwind distances.

The background, $\chi_0$, is estimated by fitting a polynomial to the measurements outside the plume. After extensive testing, the procedure described below and outlined in Fig. 3 proved to be the most effective method of filtering interplume samples. The raw $\chi$ time series is initially three-point
smoothed ($\widetilde{\chi}$) to decrease the effect of noise before further filtering (Fig. 3a). The gradient of $\widetilde{\chi}$ is then used to detect abrupt changes in $\widetilde{\chi}$ which are indicative of the UAS entering or exiting the higher-concentration plume, as shown in Fig. 3. Samples where $\nabla\widetilde{\chi}$ is greater than a specified thresh-
old get filtered out (i.e., more than ± $\gamma\sigma$, where $\gamma$ is a predefined constant; ± $0.2\sigma$ in Fig. 3b and in the subsequent analysis of measurements presented here). This gradient filter removes the majority of the inter-plume samples, leaving behind the "background" samples which vary at a much lower
frequency. However, the gradient filter is imperfect, and occasionally some inter-plume samples are not removed during this step. Therefore, a follow-up outlier filter is used to remove samples which are significantly far from the mean of the samples (more than 1 standard deviation from the mean
in Fig. 3c). The remaining samples are taken to be the background with respect to the target plume (background measurements, $\widetilde{\chi}_{bg}$), and a variable-order polynomial is fit to these samples. The polynomial order is empirically selected

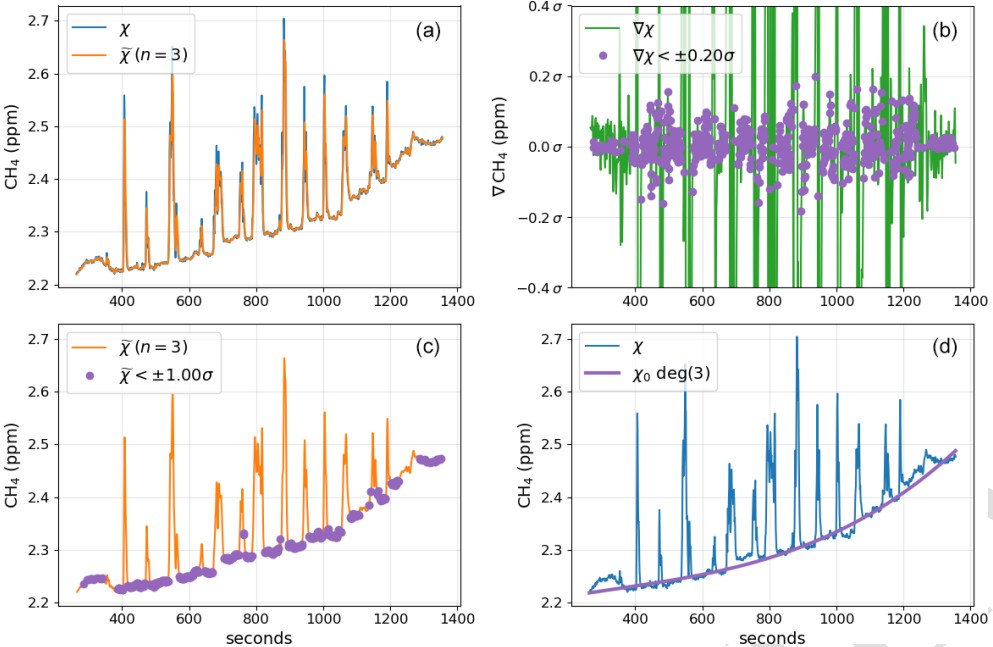

**Figure 3.** Time series of methane measurements during a typical flight, illustrating the procedure for background estimation and removal. The flight involved multiple cross-sections through the downwind plume of a known point emitter, and the plume enhancements in methane are visible as large spikes in the methane time series. **(a)** Three-point-smoothed raw data. **(b)** Gradient filter to detect the majority of in-plume samples. **(c)** Split remaining samples into segments and remove statistical outliers from each segment. **(d)** Third-order polynomial fit using remaining samples; resultant regression coefficients used with initial (non-smoothed) raw data samples used to estimate background. Estimated $CH_4$ emission rate for this dataset is $0.378 \pm 0.147 \, \text{kg h}^{-1}$ (see Sect. 2.6).

based on the individual dataset, usually between second and fourth order (cubic in Fig. 3d). Higher-order polynomials can affect the edges of the fitted time series, but generally, higher-order polynomials ($> 4$) are not required after appro-
5 priately tuning the gradient and outlier filter threshold parameters. The polynomial regression coefficients fitting $\widetilde{\chi}_{\text{bg}}$ are then applied to the original $\chi$ sampled timestamps to estimate $\chi_0$ for each independent hydrocarbon and dataset. This estimated background is then subtracted from the original
time series to get the isolated source plume enhancements, $(\chi - \chi_0)$, for both $CH_4$ and $C_2H_6$.

Figure 4 presents the distribution of $CH_4$ and $C_2H_6$ background estimate residuals ($\varepsilon_0 = \chi_0 - \widetilde{\chi}_{\text{bg}}$) from three independent flights around different source types. These three
datasets were specifically chosen to highlight the variability in $\varepsilon_0$ caused in part by the source strength and environmental conditions. The average $3\sigma$ confidence interval (99.7 %) calculated from more than two dozen flights is approximately 16 ppb for $CH_4$ and 2.5 ppb for $C_2H_6$. This minimum detec-
tion limit governs the lower bounds on source strengths and fluxes that can be quantified with this UAS (further discussed in Sect. 2.7)

## 2.4 Plume simulations

This section describes the modeling analysis that was conducted to develop optimal flight patterns and sampling strate- 25 gies for the UAS. In these idealized simulations, emissions from the target source are held constant and surrounding environmental winds are steady during the M600P's 15 to 25 min flight times. Gaussian models offer a reasonable approximation for the structure and evolution downwind of the 30 source's emission plume under these conditions (Shah et al., 2019; Stockie, 2012; Jacob et al., 2022; Seinfeld and Pandis, 2006; Meyer et al., 2022; Follansbee et al., 2024; Woodward, 1998), but it should be emphasized that source fluxes reported in this work do not rely on the results of Gaussian 35 plume simulations or inverse modeling (see Sect. 2.6).

The Gaussian plume equation for the mass density of a gas ($C$, units of $\text{kg m}^{-3}$) downwind of a point-source emitter is shown in Eq. (2).

$$
\begin{aligned}
C(x, y, z) = \frac{Q}{|\boldsymbol{u}|} &\cdot \frac{1}{2\pi \, \sigma_y(x) \, \sigma_z(x)} \\
&\times \exp\left[-y^2 / \left(2\sigma_y(x)^2\right)\right] \\
&\times \exp\left[-(z - H)^2 / \left(2\sigma_z(x)^2\right)\right]
\end{aligned} \tag{2}
$$
40

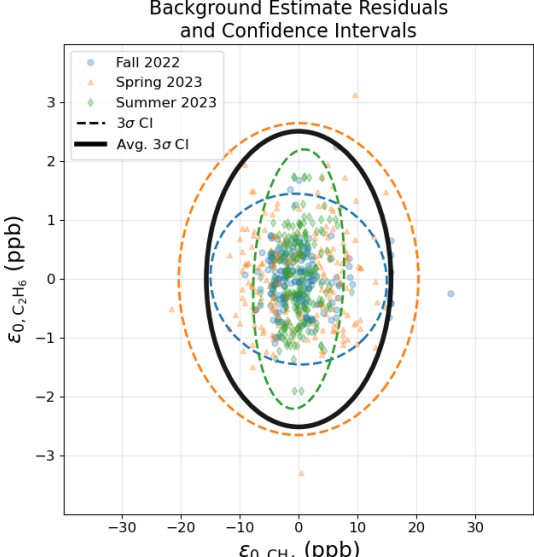

**Figure 4.** Residuals from the $CH_4$ and $C_2H_6$ background polynomial fit (outlined in Fig. 3) for three independent flights. The $3\sigma$ confidence interval (99.7 %) for each dataset is shown with a dashed line, highlighting the variability in the background estimates due to different sources and environmental conditions. The average $3\sigma$ confidence interval, calculated from 28 flights during 2022 and 2023, is overlaid with a solid ellipse.

$Q$ is the emission rate ($kg\,s^{-1}$); $\boldsymbol{u}$ is the constant horizontal wind speed ($m\,s^{-1}$), and $\sigma_z$ and $\sigma_y$ are standard deviations for Gaussian distributions in units of meters, which are generally derived using stability classes (Seinfeld and Pandis, 2006; Woodward, 1998). $H$ is the height of the source (m). $C$ is the estimated increase in gas density at position $x$ (directly downwind, centerline), $y$ (horizontal from centerline), and $z$ (vertical from centerline) given in meters. It is important to make the distinction between the mass density, $C$ ($kg\,m^{-3}$), and the mole fraction, $\chi$ ($mol\,mol^{-1}$), measured by the MIRA Pico. The conversion between the two is $C = \eta\rho\chi$, where $\eta$ is the ratio of molar masses of the gas to that of air, and $\rho$ is the air density ($kg\,m^{-3}$).

Figure 5 shows cross-sections of a Gaussian plume simulated at three downwind distances from a $25\,kg\,h^{-1}$ point source. Mean winds in this case are $\sim 2\,m\,s^{-1}$, and the stability class corresponds to high solar insolation. As expected, the simulated plume becomes more dispersed and broadens both horizontally and vertically with increasing distance from the source. Plume enhancements in methane range between 1 and 5 ppm, which are easily detectable with this system, and the horizontal and vertical scales ($\sim 50\,m$, $\sim 30\,m$) are readily accessible to the UAS.

## 2.5 Flight pattern

The full UAS setup and pre-flight checks can be completed in under 15 min, and any certified remote pilot, or person under the supervision of a certified pilot (Federal Aviation Administration, 2021), can operate and maneuver the UAS around a target source's emission plume. Part of the setup process involves the verification of MIRA measurements for both $CH_4$ and $C_2H_6$. Prior to each flight, a small canister of natural gas with a known composition ($\sim 7\,\% \; C_2H_6 : CH_4$ ) is rapidly opened and closed about 1 m upwind of the UAS gas inlet. This pre-flight release "pulse" is measured on the MIRA to test for any lag or unexpected gain offset on either of the channels. As discussed in Sect. 2.3, the raw concentration measurements can be biased due to a quasi-periodic sensor drift which must be estimated and removed before comparing the channels. This step is completed during data processing after each deployment, and it should be noted that the results from each of these controlled pulses has been consistent to within a fraction of the known $C_2H_6 : CH_4$ ratio ($< 1\,\%$ disagreement) across all deployments between 2021 and the present.

The flight time of $\sim 20\,min$ constrains the types of flight patterns that can be used to quantify source emission rates. In order to maximize the number of in-plume samples within a relatively short flight period, our strategy is to remain downwind of the source and fly horizontal "transects" perpendicular to the mean wind direction. This boustrophedonic curtain flight pattern is perpendicular to the average wind direction and involves multiple cuts through the plume in order to measure both the in-plume concentrations, $\chi$, and the ambient or background concentrations used to determine $\chi_0$. Each individual transect through the downwind plume is at a relatively constant altitude and horizontal velocity between 2 and $5\,m\,s^{-1}$, depending on the wind conditions and proximity to the source. Transects are typically 50 m to 1 km in length, depending on terrain, wind variability, and source distribution and downwind distance. Extended sources and measurements collected at larger downwind distances from the source require longer transects to ensure that the dispersing plume gets fully traversed during each transect. The ability to quickly adjust altitude between each of the individual transects allows for direct measurements of the plume's vertical structure in addition to the horizontal dispersion.

Direct measurements of a source plume with the MIRA Pico sensor require the UAS to be physically maneuvered downwind of the source. While a steady-state Gaussian plume model can be used for flux inversions (e.g., Jacob et al., 2022; Shah et al., 2019; Stockie, 2012; Seinfeld and Pandis, 2006; Woodward, 1998; Bhattacharya, 2013), these models represent plume dispersion probabilities which are not generally observed in the superposition of horizontal transects through the plume. During direct measurements, large-scale turbulence in the wind flow results in changes to the plume's size and relative location throughout the measurement period. Based on simple scaling arguments, we expect that plume variability is largest in the horizontal plane due primarily to wind directional variability. Figure 6 shows multiple simulated transects through a Gaussian plume's

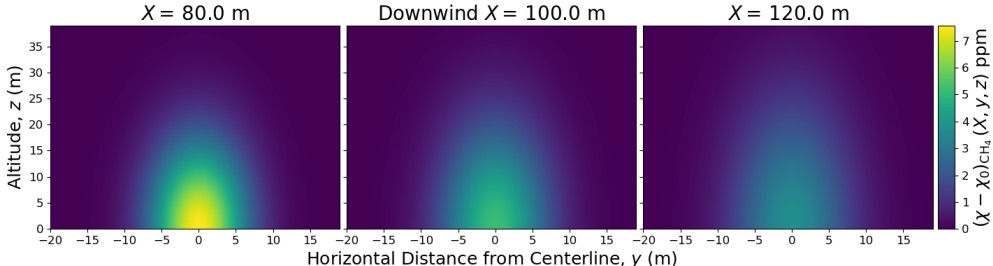

**Figure 5.** Cross-sections of modeled Gaussian plume under conditions of low wind ($2\,\mathrm{m\,s^{-1}}$) and high solar insulation (stability class "A" from Woodward, 1998). Each panel depicts the Gaussian plume cross-section at distance $X$ downwind of the source. The horizontal distance $y$ is with respect to plume centerline, and the vertical distance $z$ is above ground level. The plume model assumes a steady wind field and constant emission rate from the source.

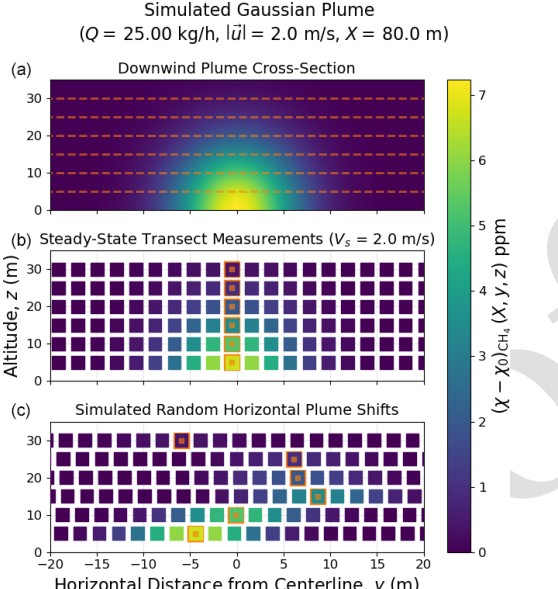

**Figure 6.** Simulated flight through Gaussian plume: **(a)** cross-section of plume $X$ meters directly downwind of source, with simulated transects shown as dashed lines. **(b)** Resample of modeled cross-section at constant horizontal transect velocity $V_s$. **(c)** Random shifts in the centerline horizontal position, simulating variable wind conditions and temporally separated plume transects.

downwind cross-section in the case of a steady-state flow. We simulate the stochastic variability in the plume by introducing random shifts in the centerline between each of the temporally separated transects, which can be seen at the bottom of Fig. 6. This pattern is generally more consistent with our observations in comparison to the steady-state model. However, the stochastic model shows that total plume flux is unaffected by horizontal plume displacements, provided that the integral is taken along horizontal transects.

## 2.6 Direct flux quantification

The data collected from a single flight are broken into $K$ individual transects through the source's downwind plume, each transect representing data collected at a nearly constant altitude, $z$, and non-zero horizontal velocity, $V_s$, while intersecting the plume. The altitude, $z$, is only adjusted between transects with the UAS outside the plume (measuring ambient or background concentrations).

Mass balance estimation techniques require knowledge of the wind speed perpendicular to the direction of travel (transect crosswind). Optimally, each transect would therefore be perpendicular to the mean horizontal wind direction ($\overline{\boldsymbol{u}}$), but local topography, changing wind fields, and flight safety considerations often result in an angle other than 90° between $\boldsymbol{V}_s$ and $\boldsymbol{u}$. Therefore, it is important to first define a unit vector, $\hat{\boldsymbol{n}}$, perpendicular to the direction of travel.

$$\begin{cases} \|\hat{\boldsymbol{n}}\| = n_x^2 + n_y^2 = 1 \\ \boldsymbol{V}_s \cdot \hat{\boldsymbol{n}} = V_x n_x + V_y n_y = 0 \end{cases} \tag{3}$$

Equation (3) shows the system of equations used to calculate this unit vector, $\hat{\boldsymbol{n}}$, using the instantaneous horizontal UAS velocities $V_x$ and $V_y$ defined previously. The transect crosswind, a measure of wind speed perpendicular to the UAS transect, is ($\boldsymbol{u} \cdot \hat{\boldsymbol{n}}$). Maintaining a constant heading and horizontal velocity during flights proved difficult during periods of irregular and shifting winds. Direct measurements of $\boldsymbol{u}$ and $\boldsymbol{V}_s$ are variable between samples, so $\hat{\boldsymbol{n}}$ is required to constrain the flux calculation with respect to each transect.

Each of the $K$ horizontal transects gets processed individually to calculate the intermediate transect-integrated flux $f_k$ (Eq. 4), the horizontal integral of the samples along each transect (units of mass flux rate per vertical distance,

$kg\,s^{-1}\,m^{-1}$).

$$f_k = \eta \sum_{i=0}^{n-1} \rho_i (\chi - \chi_0)_i (\boldsymbol{u} \cdot \hat{\boldsymbol{n}})_i \Delta s_i$$

$$= \sum_{i=0}^{n-1} (C - C_0)_i (\boldsymbol{u} \cdot \hat{\boldsymbol{n}})_i \Delta s_i \qquad (4)$$

$\eta \rho (\chi - \chi_0) = (C - C_0)$ is the background-adjusted mole fraction, $\Delta s = |\boldsymbol{V}_s| \Delta t = \sqrt{V_x^2 + V_y^2} \Delta t$ is the distance between samples ($\Delta t \approx 1\,s$), and $\rho$ and $\eta$ have been defined previously.

The total flux, $F_{\text{tot}}$, is then calculated through integrating the sum of $f_k$ and the vertical distance between physically adjacent transects, $\Delta z_k$ (Eqs. 5 and 6).

$$\Delta z_k =$$
$$\begin{cases} |\overline{z}_{k+1} - \overline{z}_k|/2 + 3|\overline{z}_k|/4 & \overline{z}_k = \min(\overline{z}_0, \ldots, \overline{z}_{K-1}) \\ |\overline{z}_{k-1} - \overline{z}_k| & \overline{z}_k = \max(\overline{z}_0, \ldots, \overline{z}_{K-1}) \\ (|\overline{z}_{k-1} - \overline{z}_k| + |\overline{z}_{k+1} - \overline{z}_k|)/2 & \text{otherwise} \end{cases} \qquad (5)$$

$$F_{\text{tot}} = \sum_{k=0}^{K-1} (f_k \Delta z_k) \qquad (6)$$

For most transects, $\Delta z_k$ ranges from the midpoint distance between the previous, lower-altitude transect up to the midpoint between the next, higher-altitude transect. Adjustments to the vertical integral step $\Delta z_k$ are taken at the bottom and top transects to account for extrapolation to the ground and above the flight pattern. The UAS cannot profile below a height of about 1.5 m, but the plume flux is expected to decrease nonlinearly to zero at the ground. Therefore, we extrapolate below the lowest flight transect by assuming a constant mixing ratio and wind speed between the lowest transect and 75 % of the distance to ground. This choice of integral quadrature corresponds to a logarithmic vertical profile for the transect-integrated flux $f$ below level $z$, $f = f_0 \ln(z/z_0)$, with $z_0 = 0.018z$. In the absence of tall trees, power lines, or other obstructions, the lowest flight transect typically lies between 2–4 m above ground level (including the UAS mast), and the assumed $z_o$ therefore ranges between about 4 and 7 cm. Extrapolation at the top edge of the plume can be more complicated, but ideally, the uppermost flight transect will lie above the plume so that a linear interpolation can adequately account for the top of the plume profile. However, some flight patterns (e.g., Fig. 9) may not completely span the possible vertical extent of the plume, and in these isolated cases we adopt a conservative approach by assuming a constant mixing ratio and wind above the transect altitude, extending only to a height defined by half the vertical distance between the top two transects. Typically, this extrapolation amounts to 2–4 m above the uppermost transect. A typical flight downwind of a natural gas point source is shown in Fig. 7. This flight is composed of 11 horizontal transects over a period of approximately 11 min. The transects were flown approximately 130 m downwind (roughly south-southeast) of the methane point source with a known emission rate of $3.42 \pm 0.01\,kg\,h^{-1}$ (El Abbadi et al., 2024).

## 2.7 Precision and uncertainties

Errors are calculated using the standard Eq. (7) described in Taylor (1996) for an arbitrary measured variable, $q$. Table 1 gives the measurement uncertainties for the major components of the UAS.

$$\delta q(x, \ldots, z) = \sqrt{\left(\frac{\partial q}{\partial x}\right)^2 \delta x^2 + \ldots + \left(\frac{\partial q}{\partial z}\right)^2 \delta z^2} \qquad (7)$$

The transect-integrated flux, $f_k$, given in Eq. (4) is a function of the background-adjusted concentration ($C - C_0$), wind speed perpendicular to UAS curtain, and the distance between transect samples ($\Delta s$). Each of these terms introduces an error in $f_k$ as defined in Eqs. (8) through (10).

The uncertainty on the background-adjusted concentration, $C$, is dependent on the calculated air density ($\rho = M_{\text{air}} P/(RT)$ $kg\,m^{-3}$), measured hydrocarbon concentration ($\chi$), estimated background ($\chi_0$; see Sect. 2.3), and the uncertainty associated with each value (see Table 1).

$$\delta(C - C_0) = \eta \sqrt{\rho^2 \delta(\chi - \chi_0)^2 + (\chi - \chi_0)^2 \delta \rho^2}$$

$$= \eta M_{\text{air}}/R \sqrt{\begin{array}{c} (P/T)^2 [\delta\chi + \delta\chi_0] + (\chi - \chi_0)^2 \\ \times [(\delta P/T)^2 + (P\delta T/T^2)^2] \end{array}} \qquad (8)$$

Note that the uncertainty on the estimated $\chi_0$ is calculated independently for each flight.

The crosswind uncertainty is dependent on the static-frame horizontal wind speed ($\boldsymbol{u}$) calculated as described in Sect. 2.2 with the full uncertainty propagation given in Appendix B. While the crosswind is dependent on the unit vector ($\hat{\boldsymbol{n}}$), the conservative upper-limit approximation for crosswind is given in Eq. (9).

$$\delta(\boldsymbol{u} \cdot \hat{\boldsymbol{n}}) = \sqrt{n_y^2 \delta u^2 + n_x^2 \delta v^2}$$

$$\approx \sqrt{\delta u(u_m, \phi, V_x)^2 + \delta v(v_m, \phi, V_y)^2} \qquad (9)$$

The final parameter in $\delta f_k$ is the separation between the sample along the transect, $\Delta s$, which is dependent on the UAS horizontal velocity ($V_h = \sqrt{V_x + V_y}$ ⁠TS1); sampling rate ($\Delta t = 1\,Hz$); and the associated uncertainties as shown in Eq. (10).

$$\delta(\Delta s) = \sqrt{\delta V_s^2 \Delta t^2 + V_s^2 \delta(\Delta t)^2}$$

$$= \sqrt{\begin{array}{c} [(V_x \delta V_x)^2 + (V_y \delta V_y)^2]^2 (\Delta t/V_s)^2 \\ + [V_x^2 + V_y^2]^2 \delta(\Delta t)^2 \end{array}} \qquad (10)$$

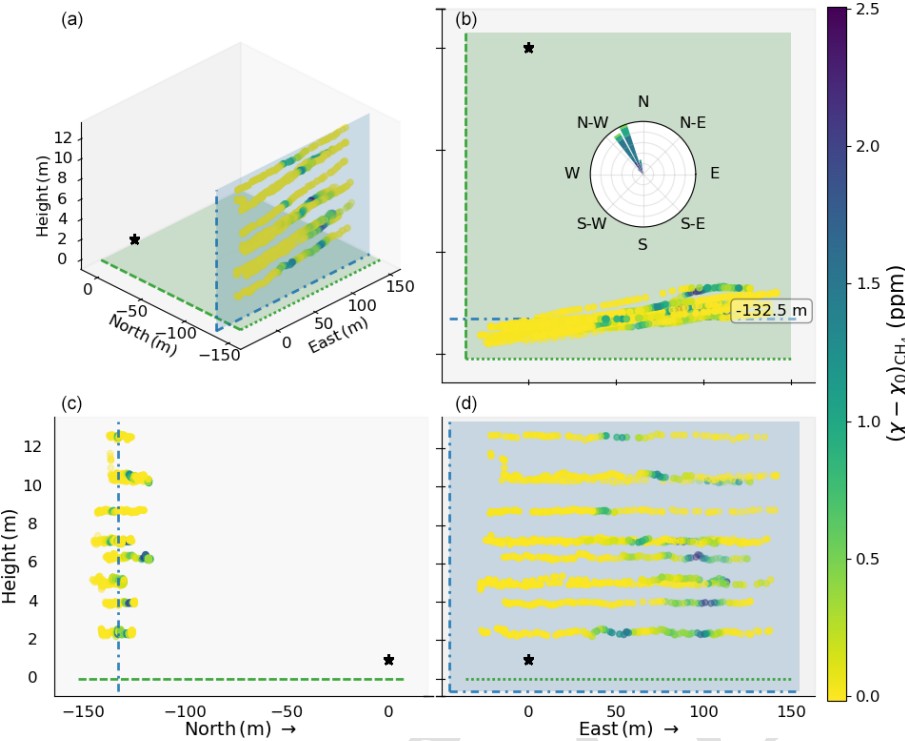

**Figure 7.** Plume transect flight pattern viewed from east **(c)**, south **(d)**, and above **(b)**. Approximate source location marked with black square. The wind rose plot **(a)** shows the Earth-fixed wind in the horizontal direction. Estimated $CH_4$ emission rate from the UAS is $4.1 \pm 1.11\,\mathrm{kg\,h^{-1}}$ (compare to metered emission rate of $3.52 \pm 0.01\,\mathrm{kg\,h^{-1}}$).

The calculated uncertainty on $f_k$ is therefore calculated as shown in Eq. (11).

$$\delta f_k{}^2 = \left[\delta(C - C_0)(\boldsymbol{u} \cdot \hat{\boldsymbol{n}})\Delta s\right]^2$$
$$+ \left[(C - C_0)\delta(\boldsymbol{u} \cdot \hat{\boldsymbol{n}})\Delta s\right]^2$$
$$+ \left[(C - C_0)(\boldsymbol{u} \cdot \hat{\boldsymbol{n}})\delta(\Delta s)\right]^2 \tag{11}$$

The total source flux is the summation of each $f_k$ multiplied by the corresponding $\Delta z_k$, as shown in Eq. (6). The vertical range of each independent transect is given in Eq. (5). $\overline{z_k}$ is the average height of all samples within transect $k$, and $\delta(\Delta z) = \sigma_{z_k}$ is the uncertainty in altitude.

$$\delta F_{\mathrm{tot}} = \sqrt{\delta f_k^2 \Delta z^2 + f_k^2 \delta(\Delta z)^2} \tag{12}$$

The measurement uncertainties detailed in Eqs. (8) through (12) can be used to identify the major sources of uncertainty in source emission rate estimates.

While the minimum flux threshold will depend on the specific meteorological conditions and flight patterns during sampling, under favorable flight conditions with high solar insulation, $2\,\mathrm{m\,s^{-1}}$ mean wind speed, and a downwind plume cross-sectional area of $100\,\mathrm{m^2}$ (see Fig. 5), the minimum source strength that can be quantified is on the order of $0.0062\,\mathrm{kg\,(CH_4)\,h^{-1}}$ and $0.0018\,\mathrm{kg\,(C_2H_6)\,h^{-1}}$.

## 3 Results

This section gives an overview of a number of quantification flights between spring 2022 and fall 2023. Section 3.1 details the results of multiple controlled releases using the system described in El Abbadi et al. (2024). Section 3.2 discusses measurements from various, smaller biogenic and thermogenic emission sources collected between winter 2021 and fall 2023.

### 3.1 Controlled-release experiments

In fall of 2022, the NMT team participated in a single-blind controlled-release validation campaign in Casa Grande, Arizona, USA (Sherwin et al., 2024; El Abbadi et al., 2024). While this validation campaign focused on larger-scale aircraft and satellite system validation, it was a unique opportunity to quantify the accuracy for this method of direct methane flux quantification. The UAS was deployed multiple times around the controlled-release location during two separate 3 d validation trips: 10–12 October and 14–16 November 2022. Multiple flights at various times throughout the daily testing window (between 10:00 and 14:00 MST most days) measured concentrations downwind of the release stack.

Suboptimal environmental conditions and unforeseen instrumentation issues resulted in the rejection of the major-

ity of October 2022 flights from the analysis process. The November 2022 3 d campaign produced a total of 12 flights, 8 of which were during times of low-variability wind fields and good flight conditions to allow for reliable flux quantification.

It is important to note that, at the time of this campaign, the UAS flight pattern and quantification method were still in active development. While the NMT team quantified and submitted flux estimates during the single-blind and partially blinded phases of the unblinding process (described in El Abbadi et al., 2024), the final unblinded metered emissions were invaluable to debug and update the initial quantification process, ultimately leading to the numerical integration outlined in Sect. 2.6. Here, we present the analysis of the quantification results informed by the unblinded, ground-truth methane flow rates.

Figure 8 shows the comparison between UAS-calculated flux and the corresponding unblinded metered emissions from eight independent November 2022 flights. The metered rates, taken to be the ground-truth emission rates, range from 1.7 to $1500 \, \mathrm{kg\,h^{-1}}$. This broad range of emission rates highlights the system's dynamic range, and the fitted linear regression shows the quality of emission estimates compared to the one-to-one agreement with metered rates (dotted line). Our method of flux quantification shows reasonable agreement with the expected emission rates above $1 \, \mathrm{kg\,h^{-1}}$. However, the results suggest an underestimation of the emissions which is not correlated to number of transects or wind speeds during flights. This may be due to the limited flight time and under-sampling the downwind plume with the UAS. One interpretation of these results is that our field measurements could represent a reasonable lower bound to the true emission rate of the source.

## 3.2 Targeted point and distributed sources

The deployment strategy for this UAS is designed to quantify emissions from targeted local sources such as O&NG wells, manure storage, and biogenic lagoons. With a limited operational distance and flight time, larger distributed sources (such as large-scale dairies and agricultural centers) are difficult to properly quantify using the numerical integration technique described in Sect. 2.6. However, smaller-scale municipal waste facilities with heterogeneous emission profiles can be quantified as long as the UAS is able to fully, and repeatedly, transect the complex and irregular plumes downwind of the facility.

### 3.2.1 Municipal waste facility

During the course of system development and testing, the UAS was deployed around a local municipal waste facility (MWF) in Socorro, New Mexico (NM). This location serves a county population of approximately 16 300 with multiple cells over an area of 45 ha. Larger facilities studied by

Olaguer et al. (2022) and Lan et al. (2015) reported emission rates between 85 and $> 2000 \, \mathrm{kg\,h^{-1}}$; this much smaller local facility, however, was expected to emit at rates less than $10 \, \mathrm{kg\,h^{-1}}$ (Olaguer et al., 2022; Bogner and Matthews, 2003). The low emission rate and temporal heterogeneity, likely due to changes in cell activity, are evident across the multiple visits between spring 2022 and summer 2023. The flight path from one such deployment can be seen in Fig. 9.

In comparison to flights downwind of point sources such as O&NG wells or controlled natural gas (NG) releases (Fig. 7), the flight pattern in Fig. 9 was positioned about 0.5 km downwind of the source and involved plume transects up to $\sim 500 \, \mathrm{m}$ long, and over an altitude range of up to 50 m. In addition, the methane enhancements are about a factor of 10 lower than in Fig. 7, and the plume is distributed over a larger area and centered at a higher altitude. Nevertheless, the overall plume structure allows for the same flux analysis as discussed for the localized NG point sources above. The measured methane emission rate for this flight was $1.33 \pm 0.58 \, \mathrm{kg\,h^{-1}}$.

### 3.2.2 Orphan well

During April 2023, the UAS was also deployed around an "orphaned well" located in Hobbs, New Mexico, USA. This site has been out of use for more than 2 decades and was in the initial phase of being plugged when visited in April 2023. A detailed analysis of data from this field campaign is presented in Follansbee et al. (2024). The observed plume structure was similar to that shown in Fig. 7, but mean wind speeds were much larger and methane plume enhancements were $\sim 400 \, \mathrm{ppb}$, roughly between the range of plume enhancements shown in Figs. 7 and 9. Three downwind flights yielded consistent fluxes in the range of 0.3 to $0.4 \, \mathrm{kg\,h^{-1}}$ (Follansbee et al., 2024).

### 3.2.3 Wastewater treatment plant

Wastewater treatment plants (WWTPs) are a known source of biogenic methane emissions via biodegradation of pollutants by anaerobic bacteria (Song et al., 2023). The small town of Socorro, NM, has a local wastewater treatment plant that processes less than $1 \times 10^6 \, \mathrm{gal\,d^{-1}}$ ($44 \, \mathrm{L\,s^{-1}}$), so its contribution to anthropogenic methane is relatively quite low (when compared to O&NG and agricultural operations). However, the low emission rate of this location was useful for the testing of the UAS's lower detection and quantification limits.

A time series of measured $CH_4$ and $C_2H_6$ concentrations from a short flight in August 2023 is shown in Fig. 10. The $CH_4$ and $C_2H_6$ detection limit (calculated from the background fit residual $3\sigma$ confidence interval; see Sect. 2.3) is overlaid on the time series shown in Fig. 9. While the in-plume $CH_4$ concentration levels peak at around 40 ppb, no corresponding $C_2H_6$ enhancements are detected while pass-

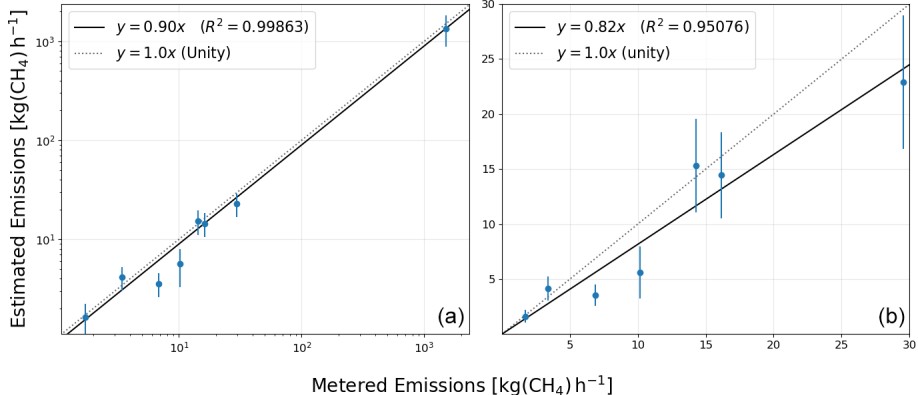

**Figure 8.** Comparison of the UAS $CH_4$ flux calculation versus metered emissions reported by El Abbadi et al. (2024) and Sherwin et al. (2024). Both plots show eight individual flights during the November 2022 field campaign with $2\sigma$ error bars. Linear regression fit shown as solid line for comparison with the idealized one-to-one fit corresponding to perfect agreement between UAS and metered data (dotted line). **(a)** Logarithmic plot highlighting the UAS's large dynamic range. **(b)** Linear plot focused on emission rates below detection limits of most aircraft quantification methods; linear regression fit does not include emission rates above $100\,\mathrm{kg\,h^{-1}}$.

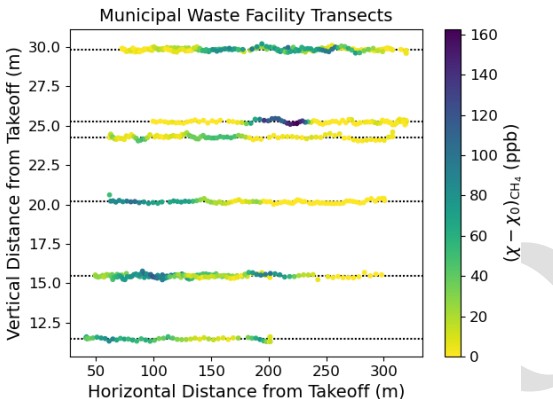

**Figure 9.** Example flight, split into individual transects, from a landfill in New Mexico. The landfill itself is composed of active and inactive cells distributed over an area of approximately 45 ha, resulting in a non-uniform downwind plume profile. In comparison to the localized "point source" presented in Fig. 7, longer horizontal transects across a wider vertical range are required to measure the multiple distributed and highly variable emitters. Estimated flux rate for this flight is $1.33 \pm 0.58\,\mathrm{kg\,h^{-1}}$.

ing through the plume. The lack of a correlation between $C_2H_6$ and $CH_4$ is strong evidence that the measured plume is from anaerobic digestion at the WWTP and is not contaminated by another source. However, it is important to note that weaker source strengths lead to a decrease in the signal-to-noise ratio, so $C_2H_6$ from thermogenic sources may not be detectable. This is shown visually at the bottom of Fig. 10 where the expected $C_2H_6$ time series for 10 % and 5 % $C_2H_6$ : $CH_4$ thermogenic plumes are modeled with respect to the measured $CH_4$ time series. While the $C_2H_6$ signal for the 10 % thermogenic mixture peaks above the detection threshold (1.5 ppb in this dataset), the 5 % mixture

is barely visible above the noise floor. However, it is important to note that weaker source strengths lead to a decrease in the signal-to-noise ratio, so $C_2H_6$ from thermogenic sources may not be detectable. This is shown visually at the bottom of Fig. 10 where the expected $C_2H_6$ time series for 10 % and 5 % $C_2H_6$ : $CH_4$ thermogenic plumes are modeled with respect to the measured $CH_4$ time series. While the $C_2H_6$ signal for the 10 % thermogenic mixture peaks above the detection threshold (1.5 ppb in this dataset), the 5 % mixture is barely visible above the noise floor. The emission rate for the flight shown in Fig. 10 is estimated to be $68.7\,(+171.5, -61.7)\,\mathrm{g\,h^{-1}}$.

## 3.3 Source attribution

As discussed in Sect. 1, simultaneous measurements of both $CH_4$ and $C_2H_6$ are useful for source classification as either biogenic or thermogenic. In addition, there are varying ratios of trace hydrocarbons found in thermogenic natural gas sources, and $C_2H_6$ is the second-most dominant compound in refined natural gas (Peischl et al., 2018; Hodnebrog et al., 2018; Kutcherov and Krayushkin, 2010; Glasby, 2006; Space Studies Board et al., 2019; Solomon et al., 2007; Hansen et al., 2000; Meyer et al., 2022). Therefore, the ratio of $C_2H_6$ to $CH_4$ can be used to estimate the percentage of non-methane compounds in an O&NG plume and distinguish between sources.

Figure 11 shows the ratio of measured $C_2H_6$ and $CH_4$ mole fraction from five different field measurements around various sources, including the municipal waste facility discussed in Sect. 3.2.1. Multiple downwind plume measurements showed a negligible $C_2H_6$ content ($< |0.2\,\%|$ $C_2H_6$ : $CH_4$) from this biogenic source. Consistent with measured concentrations in Fig. 9, $CH_4$ concentrations from the MWF

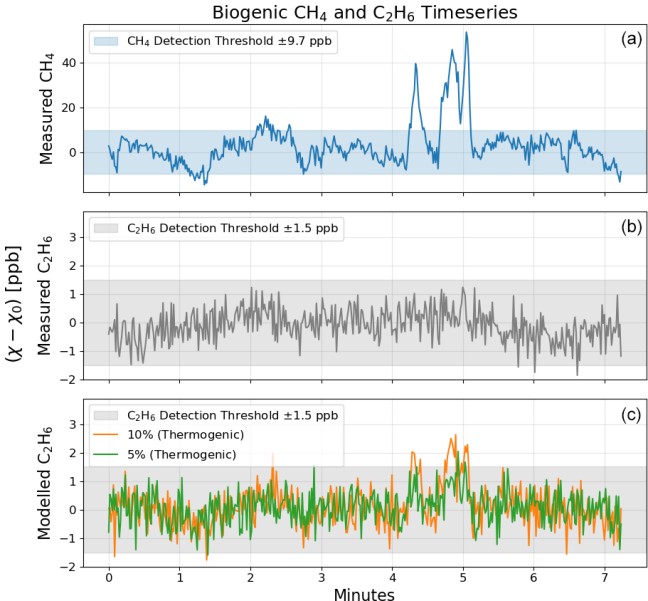

**Figure 10. (a, b)** Time series of measured and background subtracted $CH_4$ and $C_2H_6$ concentrations (Sect. 2.3) from a flight downwind of the city of Socorro wastewater treatment plant in August 2023. **(c)** Expected $C_2H_6$ time series if measured $CH_4$ plume (top) was thermogenic with 10 % and 5 % $C_2H_6$ : $CH_4$ ratios. The mean winds were $\sim 5\,\mathrm{m\,s^{-1}}$, and the solar insolation corresponded to stability class "B" due to partial cloud coverage (Seinfeld and Pandis, 2006; Woodward, 1998). The plume was intercepted on two transects separated by about 40 s, as seen in the $CH_4$ time series (top) about 4–5 min into the flight. No corresponding $C_2H_6$ enhancements we measured indicated that the plume is biogenic (compare with expected thermogenic $C_2H_6$ time series). Estimated $CH_4$ emission rate of $0.061 \pm 0.032\,\mathrm{kg\,h^{-1}}$.

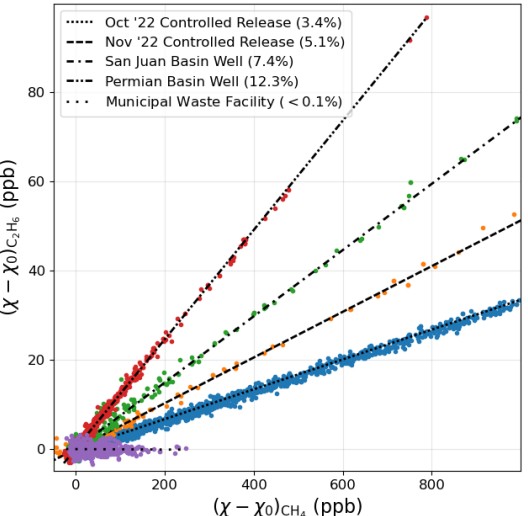

**Figure 11.** Ethane and methane enhancement ratios from multiple independent field tests at different sites. "October and November 2022 controlled releases" are two independent multiday field campaigns in Casa Grande, AZ, during a controlled NG release. "San Juan Basin well": multiple flights near an active wellhead in Cuba, NM. "Permian Basin well": three independent flights downwind of an orphan well in Hobbs, NM. "Municipal waste facility" is a relatively small waste disposal location near Socorro, NM.

are much lower in comparison to the sampled thermogenic methane sources.

The two different controlled-release campaigns in fall 2022 were about 1 month apart, and these data are segregated in Fig. 11. The measured percentages are in agreement with the contemporaneous $CH_4$ concentration measurements reported by the controlled-release team (El Abbadi et al., 2024). The same ratio of $C_2H_6$ to $CH_4$ was measured across all flights during the first release and similarly during the second campaign, though the ethane content was noticeably different during the second campaign.

Emissions of unrefined NG from the orphan well discussed in Sect. 3.2.2 contained a smaller fraction of $CH_4$ due to the presence of other compounds such as $H_2S$, so it is incorrect to approximate the $C_2H_6$ ratio in Fig. 11 as the non-$CH_4$ percentage. However, this unique $C_2H_6$ : $CH_4$ ratio was consistent across all orphan well flights and in good agreement with ground-based systems deployed at the same site (Follansbee et al., 2024); the larger ethane content is due to the unrefined natural gas seeping from the unused well (Stolper et al., 2018). Furthermore, measurements of unrefined NG

from a leaking O&NG well in the San Juan Basin, obtained in October 2023, showed a distinct ratio from that of the orphan well from the Permian Basin. The San Juan Basin is primarily a coal-producing region and therefore has a noticeably lower $C_2H_6$ content compared to the Permian Basin, the latter region being primarily composed of oil. Thus, each of the sources shown in Fig. 11 had a unique and consistent $C_2H_6$ ratio which can be used to characterize and differentiate multiple sources based on this percentage. This is in agreement with the findings of Meyer et al. (2022) and Germain-Piaulenne et al. (2024).

Figure 12 highlights the dynamic range of the UAS, which is able to detect $CH_4$ enhancements of as little at $20\,\mathrm{ppb\,s^{-1}}$ (after background removal; Sect. 2.3). Additionally, the UAS has demonstrated the ability to measure much larger concentrations of more than 40 000 ppm (empirically determined based on MIRA Pico saturation levels during Arizona (AZ) validation flights).

## 4   Conclusions

Our results demonstrate the capabilities of this integrated UAS package, along with associated flight strategies and data analysis methodologies, to quantify and characterize methane of point and distributed emission sources. Through direct, in situ measurements of $CH_4$ concentrations and vec-

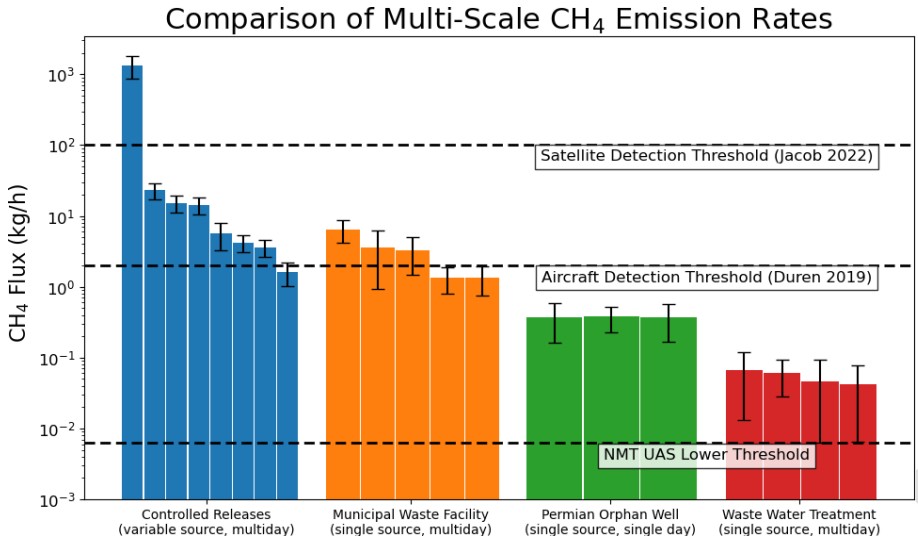

**Figure 12.** Comparison of estimated methane flux from various anthropogenic sources. The UAS has a large dynamic range capable of quantifying emission rates from smaller sources (less than $1\,\mathrm{kg\,h^{-1}}$). Horizontal dashed lines show the absolute lowest detection limits for satellite and aircraft quantification methods (Jacob et al., 2022; Duren et al., 2019) with respect to the estimated lower limit of this system ($\sim 0.007\,\mathrm{kg\,h^{-1}}$; see Sect. 2.7).

tor winds downwind of target sources, the UAS can be used to quantify emission rates spanning more than 3 orders of magnitude. The lower-bound limit for $CH_4$ flux measurements is estimated to be $\sim 0.007\,\mathrm{kg\,h^{-1}}$, and it is determined primarily by the precision of the concentration measurements and uncertainties associated with determining the background levels with respect to variable plume enhancements.

Environmental conditions largely dictate the plume dispersion and transport with respect to a target source; optimal downwind distances and transect velocities during flight are highly dependent on insolation and wind speed and direction. The results from the controlled releases shown in Fig. 8 and Table B1 were obtained during daytime periods of relatively steady winds and low cloud coverage (high solar insulation leading to large vertical mixing). In all cases, the measured emission rates showed reasonable agreement with metered flow rates, generally within $1\sigma$ error. However, the results suggest the potential for a small systematic underestimation of emissions across all scales, depending on environmental conditions. This underestimation may arise from limitations on the horizontal extent or vertical distribution of transects during each flight, resulting in an under-sampling of the continuous emission plume. Therefore, UAS flux quantification may represent an effective lower-bound value for the true emission rate. It is important to note that the results in Sect. 3.1 were informed by the fully unblinded metered rates, whereas the aforementioned aircraft systems used fully blinded or partially unblinded data for the reported estimates (El Abbadi et al., 2024; Sherwin et al., 2024).

The primary limitations of our UAS measurement approach are related to meteorological conditions. Optimum mean wind speeds are in the range of $2$–$6\,\mathrm{m\,s^{-1}}$. Wind speeds below $2\,\mathrm{m\,s^{-1}}$ have been shown to produce less reliable fluxes due to higher variability in the plume position and shape, while winds above $6\,\mathrm{m\,s^{-1}}$ are too strong for safe flights with this UAS. The steadiness of wind directionality can also be a factor, although it is a much less severe constraint than speed provided that the flight pattern is sufficiently wide to intersect the plume on every transect. Proper deployment also requires access for takeoff and landing within $1\,\mathrm{km}$ of the source, and the target source's location must also be known to within a few kilometers. While successful deployment and flux quantification is constrained to a specific range of environmental conditions, the UAS offers a relatively simple method of deployment for quantifying emitters that would otherwise be difficult or impossible for other monitoring techniques to access or detect. Thus, although this system is not optimized for wide-area surveys, it is well suited for site quantification of known sources such as O&NG wells or processing facilities, small dairies, municipal waste facilities, and wastewater treatment plants.

## Appendix A:  Supplemental information

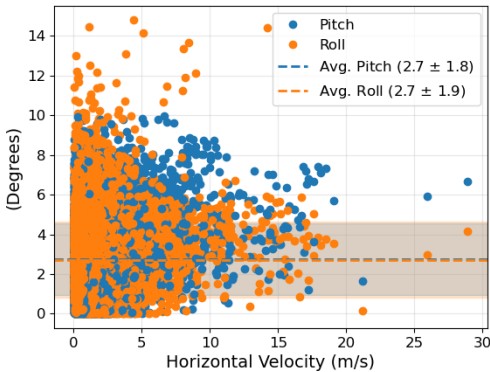

**Figure A1.** Instantaneous UAS pitch and roll versus horizontal velocity from all flights presented in Fig. 12 and Table B1 (> 13 500 samples). Plot contains all samples prior to filtering and transect processing as described in Sect. 2.6.

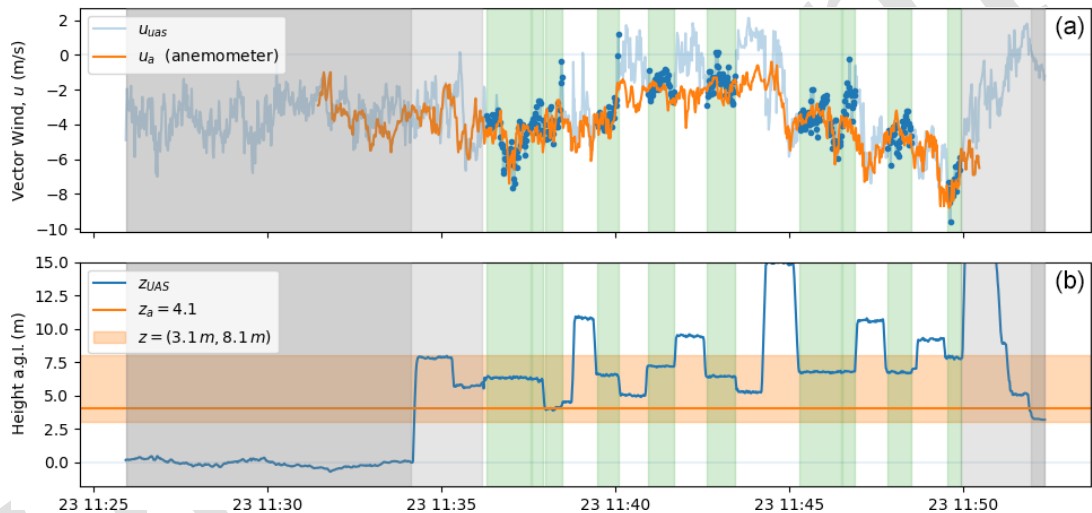

**Figure A2. (a)** Comparison of vector wind ($u$) from UAS adjustment (Sect. 2.2) with directional ground anemometer aligned to the west. **(b)** Height of UAS above ground level (blue) with respect to anemometer (orange). Highlighted regions depict steady and level transects within 20 m horizontal distance from the fixed anemometer.

## Appendix B:  Supplemental equations

### Static-vector wind uncertainty

$$\delta u = \sqrt{\left(\frac{\partial u}{\partial u_m}\right)^2 \delta u_m{}^2 + \left(\frac{\partial u}{\partial \phi}\right)^2 \delta \phi^2 + \left(\frac{\partial u}{\partial V_x}\right)^2 \delta V_x{}^2} \quad \text{(B1)}$$

$$\delta v = \sqrt{\left(\frac{\partial v}{\partial v_m}\right)^2 \delta v_m{}^2 + \left(\frac{\partial v}{\partial \phi}\right)^2 \delta \phi^2 + \left(\frac{\partial v}{\partial V_y}\right)^2 \delta V_y{}^2} \quad \text{(B2)}$$

$$\frac{\partial u}{\partial u_m} = \cos(\phi), \ \ \frac{\partial u}{\partial \phi} = -u_m \sin(\phi) + v_m \cos(\phi), \ \ \frac{\partial u}{\partial V_x} = 1 \quad \text{(B3)}$$

$$\partial v \partial v_m = \cos(\phi), \ \ \frac{\partial v}{\partial \phi} = -v_m \sin(\phi) - u_m \cos(\phi), \ \ \frac{\partial v}{\partial V_y} = 1 \quad \text{(B4)}$$

**Table B1.** Measured emission rates from various source types shown in Fig. 12. Presented error ranges are the $3\sigma$ bounds calculated from sensor uncertainties and background estimation as described in Sect. 2.6.

| Source type | $F_{tot}$ [CH$_4$] (kg h$^{-1}$) | | $F_{tot}$ [C$_2$H$_6$] (kg h$^{-1}$) | | $\langle|\boldsymbol{u}|\rangle$ (m s$^{-1}$) | $\langle\angle\boldsymbol{u}\rangle$ (deg) | $\langle T\rangle$ (C) | $\langle P\rangle$ (hPa) | $K$ | Duration (min) | Date |
|---|---|---|---|---|---|---|---|---|---|---|---|
| Controlled release | 1342.624 | [875.334, 1809.914] | 25.831 | [16.888, 34.774] | 2.6 | 391.1 | 12.7 | 969.9 | 10 | 11:24 | 15 Nov 2022 |
| Controlled release | 22.904 | [16.862, 28.946] | 1.475 | [1.044, 1.906] | 4.3 | 318.9 | 11.2 | 974.8 | 7 | 12:46 | 16 Nov 2022 |
| Controlled release | 15.300 | [11.060, 19.540] | 0.962 | [0.681, 1.243] | 4.5 | 363.9 | 15.1 | 974.2 | 7 | 14:00 | 16 Nov 2022 |
| Controlled release | 14.438 | [10.498, 18.378] | 0.876 | [0.617, 1.135] | 3.4 | 325.5 | 16.8 | 968.0 | 8 | 13:03 | 15 Nov 2022 |
| Controlled release | 5.630 | [3.261, 7.999] | 0.345 | [0.192, 0.498] | 1.6 | 334.2 | 11.4 | 970.6 | 12 | 16:59 | 15 Nov 2022 |
| Controlled release | 4.154 | [3.040, 5.268] | 0.264 | [0.168, 0.360] | 5.1 | 333.5 | 12.8 | 974.9 | 11 | 17:59 | 16 Nov 2022 |
| Controlled release | 3.564 | [2.606, 4.522] | 0.218 | [0.124, 0.312] | 4.6 | 356.2 | 13.8 | 974.9 | 6 | 10:45 | 16 Nov 2022 |
| Controlled release | 1.629 | [1.035, 2.223] | 0.100 | [0.057, 0.143] | 3.5 | 333.8 | 15.3 | 968.8 | 15 | 18:44 | 15 Nov 2022 |
| Municipal waste facility | 6.476 | [4.187, 8.765] | | – | 3.8 | 354.2 | 19.7 | 862.7 | 10 | 12:19 | 31 Aug 2022 |
| Municipal waste facility | 3.625 | [0.944, 6.306] | | – | 4.8 | 337.9 | 18.9 | 862.7 | 8 | 11:19 | 31 Aug 2022 |
| Municipal waste facility | 3.279 | [1.473, 5.085] | | – | 7.1 | 326.8 | 26.1 | 858.2 | 6 | 11:56 | 10 Aug 2023 |
| Municipal waste facility | 1.336 | [0.806, 1.866] | | – | 3.5 | 324.0 | 24.2 | 856.6 | 6 | 15:29 | 29 Jun 2023 |
| Municipal waste facility | 1.331 | [0.744, 1.918] | | – | 5.5 | 318.3 | 7.6 | 867.5 | 6 | 11:13 | 28 Oct 2022 |
| Permian orphan well | 0.374 | [0.165, 0.583] | 0.085 | [0.021, 0.149] | 9.4 | 218.9 | 22.1 | 886.1 | 4 | 05:19 | 19 Apr 2023 |
| Permian orphan well | 0.378 | [0.230, 0.526] | 0.105 | [0.067, 0.143] | 5.9 | 182.6 | 16.7 | 886.7 | 11 | 11:44 | 19 Apr 2023 |
| Permian orphan Well | 0.368 | [0.170, 0.566] | 0.089 | [0.038, 0.140] | 5.8 | 209.3 | 17.7 | 886.7 | 7 | 09:29 | 19 Apr 2023 |
| Wastewater treatment | 0.066 | [0.013, 0.119] | | – | 2.1 | 230.0 | 9.5 | 855.9 | 8 | 10:14 | 28 Feb 2023 |
| Wastewater treatment | 0.061 | [0.029, 0.093] | | – | 4.6 | 136.4 | 24.2 | 862.2 | 3 | 07:14 | 31 Aug 2023 |
| Wastewater treatment | 0.046 | [0.006, 0.093] | | – | 4.2 | 136.6 | 24.0 | 862.5 | 7 | 11:50 | 31 Aug 2023 |
| Wastewater treatment | 0.042 | [0.007, 0.077] | | – | 7.0 | 152.8 | 20.6 | 860.8 | 13 | 12:15 | 6 Jun 2023 |

*Code and data availability.* Software minimal working examples (MWEs) for MIRA Pico background concentration estimation (Sect. 2.3) and static wind calculations using measurements with the onboard anemometer (Sect. 2.2) are available through GitHub (https://github.com/jfdoolster/nmt_uas_background_estimate.git, Dooley, 2024a; https://github.com/jfdoolster/nmt_uas_wind_direction.git, Dooley, 2024c). Processed dataset files from flights discussed in Sect. 3, Fig. 12, and Table B1 are also available on GitHub (https://github.com/jfdoolster/nmt_uas_public.git, Dooley, 2024b).

*Author contributions.* JFD and KM designed and fabricated the UAS. JFD conducted field tests and measurements and developed the analysis algorithms with guidance and assistance from KM. MKD and AGM assisted in the initial system design and flight planning. SHEA and EDS designed and managed the methane controlled-release campaigns. MKD, EF, and JEL assisted with site access and data collection and interpretation. The manuscript was written primarily by JFD and KM, with important contributions from all coauthors.

*Competing interests.* The contact author has declared that none of the authors has any competing interests.

ther geographical representation in this paper. While Copernicus Publications makes every effort to include appropriate place names, the final responsibility lies with the authors.

*Acknowledgements.* Support for instrumentation and UAS equipment provided in part by the New Mexico Tech Office of Research. This project has benefited from collaborations with the New Mexico Bureau of Mines, the New Mexico Petroleum Recovery Research Center, and Los Alamos National Laboratory. We thank the NMT Facilities Management Office for their support of field measurements during the course of this project.

Special thanks to operators of the controlled release in Casa Grande, AZ, recognizing the team members who contributed to field operations during the October and November 2022 testing periods and to processing the ground-truth data that were provided (in alphabetical order by last name): Adam R. Brandt, Philippine M. Burdeau, Yuanlei Chen, Zhenlin Chen, Jeffrey S. Rutherford, and from Rawhide Leasing (operating the gas equipment) Mike Brandon.

*Financial support.* Support for the UAS instrumentation and equipment was provided by an internal seed grant through the President's Office at New Mexico Tech. Additional funding support for operations and analysis was provided by the New Mexico Consortium, subaward (grant no. 043699).

*Review statement.* This paper was edited by Glenn Wolfe and reviewed by two anonymous referees.

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

**Remarks from the typesetter**

TS1    For this change we need the approval from the editor. Please provide a statement why this needs to be changed.

TS2    Please provide your last access date.