# Peer review of "A New Aerial Approach for Quantifying and Attributing Methane Emissions: Implementation and Validation"

_EGUsphere, 2024_

## Referee Comment (RC1)

**Review of "A New Technique for Airborne Measurements to Quantify Methane Emissions Over a Wide Range: Implementation and Validation" – Dooley et al.**

**General comments:**

The manuscript describes the use of an Unmanned Aerial System, equipped with a fast-response methane and ethane sensor and ultrasonic anemometer, for estimating methane emissions from various point sources. The UAS payload configuration, sampling and flight, and analysis and verification techniques are outlined in the manuscript alongside estimates of methane emissions from various point sources, including controlled releases. Due to the availability of coincident ethane and methane measurements, ratios of ethane to methane enhancements above background levels are also evaluated for these various point sources. Overall, there is a need to bridge the gap between aircraft and/or satellite-based emissions estimates and ground-based, bottom-up estimates (and limitations therein for both). The novel use of UAS to provide the capability to bridge spatial and detectability gaps is important to describe for estimating emissions from smaller or distributed leaks and distinguishing biogenic versus anthropogenic sources. Overall, a great deal of testing has been done with this system and onboard payload and I would recommend publication in AMT after addressing the comments below that highlight places where clarifications to the manuscript structure to improve readability.

**Specific comments:**

The title is a bit vague in referencing a "wide range" - it is referenced that the UAS can sample scales of up to 1 km, but aircraft can sample at much wider ranges... would "A new UAS-based technique for quantifying and attributing methane emissions from small and distributed point sources: [...]" or similar be more relevant?

- L294 and L376 mention a limited operational distance and flight time being potentially prohibitive, so these comments also support changing the title to be more descriptive of the technique and its capabilities.

In general, the manuscript is organized appropriately, but structuring within sections could be either reconfigured, renamed, or provide more detail. In general, it could be helpful to add a traditional "Methods" section, which encompasses Sections 2-3, whereas "Results" can include section 4.

- When first describing the system and onboard payload, these sections jump around a bit, with 2.2.1 seeming a bit out of place (and perhaps fitting better under 2.1, Onboard sensors). Consider changing for clarity.
- The information in L110-116 seems more like "onboard data logging and transmission"
- Section 2.5: please explain why this section is needed; L193-195 provides the motivation for this paragraph, but this motivation (and perhaps L196-203) should be described and presented at the beginning of this section.
- Section 4: this section is difficult to follow with the many flights that occurred at different times, with varying purposes (e.g. October/November 2022 controlled release experiments; Socorro, NM MWF, Spring 2022-Summer 2023; Orphan Well, April 2023;

WWTP Summer 2023. A table outlining the flights, locations, or purpose, or even more descriptive section headers and a few sentences describing each might help. To me, the distinction between (a) controlled-release experiments and validation of the system and (b) smaller, point source emissions detection via case studies is important.

• L274-276 can be moved to the methods as this describes the flight strategy.

Background determination: L165-169 could be expanded upon to detail the procedure that is plotted in Figure 3. Please also describe the gradient method you use in 3b) within the text (and not just the caption) and how this is used in 3c).

- Figure 7: This can be described more effectively, in general. If I am reading this correctly, this is the uncertainty in the background estimation from just two test flights, which ties to Section 2.4 and is better explained prior to the uncertainty estimate in Section 3.3. To me, this figure and its description would be better suited just following Figure 3, where readers can directly connect the uncertainty in background CH4 to CH4 emissions rate uncertainties.
- Do you have estimates of how results in Figure 7 compare to all flights (i.e. are the two test flights representative of typical flights)? Why is the baseline uncertainty not incorporated on a per-flight basis?
- The word "baseline" is used interchangeably with "background" and it would be clearer in the text to just use one or the other.

Section 3.3: this section needs to be expanded upon so that it is clear where uncertainties in each term in Equations 4-6 come from and how they contribute to the overall uncertainty in  $F_{tot}$ . For example, Table 1 seems to only describe onboard UAS sensor precision and/or accuracy, but not include other sources of uncertainty like the uncertainty in CH4 and C2H6 enhancements, or uncertainties in (u \*n). All of these propagated uncertainties should be incorporated in the lower LOD of the flux estimate, correct?

- L96: wind speed magnitude uncertainty is 0.35 m/s, but Table 1 states 0.2 m/s. Please describe how the wind speed uncertainty is derived.
- The MIRA CH4 uncertainty is stated as 10 ppb here, but the error in background derivation is stated as 20 ppb. Is the propagation of MIRA CH4 precision and the background uncertainty taken into account in Ftot, which likely adds to the overall uncertainty and lower LOD? It is unclear how the background uncertainty from each flight is incorporated into the total flux uncertainty. All of this would be very advantageous to outline in Section 3.3, and similarly, for C2H6.
- It should be stated somewhere how the MIRA is calibrated prior to each flight for CH4 and C2H6
- Please state what is meant by "standard flight conditions"

The uncertainties for each of the various source emission rates calculated in section 4 should be stated alongside the measured emissions rates. Some are stated, but others are not (e.g. in 4.1.2). This is important when assessing how well the technique might fare with one type of source versus another... Why is the upper uncertainty in 4.1.3 for the WWTP up to 250%, and why does the lower uncertainty differ? Because this is an AMT manuscript, it would benefit the reader to

offer explanations for the calculated uncertainties in each of the various cases to assess limitations and capabilities of this technique.

L289: The results of the controlled release experiments indicate that there is a "systematic" underestimation, but this does not look systematic as some estimates are higher than the metered emissions rates. Can the setup of the experiment be explained in more detail here? The type of point source is mentioned, with wind conditions, but how was the UAS flown and does this contribute to this underestimation? There are no hypotheses provided for why emissions estimates from the UAS system are lower in general – can you provide some?

**Technical comments:**

- Abstract and throughout: "UAS" is used, but also "UAV" please choose one or the other to be consistent
- Figure 1, caption, and L126: What does "dual-opening" mean with respect to the sampler inlet?
- L84: precision on CH4 and C2H6 is not what is specified in the abstract please confirm which is correct.
- L86: below, the response time was ~ 2 s please state one or the other for consistency
- L90: mole fractions are presented here in ppm or ppb, which is a mol/mol. Please correct either mixing ratio or mole fraction determinations throughout.
- L141: symbol for 'yaw' different here than it is in Eq. 1.
- L145: Please punctuate and define Eq. 1
- L173: ...as shown in 'Figure' 3
- L174: Sentence fragment starting at '+\-'
- Equation 2: please describe as Eq 2 in text and punctuate.
- Equation 3: same as above
- Equation 4: please punctuate within paragraph
- Equations 5 and 6: same as above
- L189: 'volume mixing ratio' as described in mol/mol is actually a mole fraction
- L194: 'ppmv' is described here but CH4 is described in ppm elsewhere
- Figures 11 and 12: please correct spelling of municipal
- Figure 12: What do the individual colors mean? If nothing, does it make sense to have controlled release vs. municipal vs. orphan vs. WWTP all be different colors?
- Figure 6: X-Xo is used to show an enhancement, whereas in L338, a delta symbol is used, in addition to both being used in Figure 10 – please choose one or the other for clarity.
- L283: please change ground truth-methane to 'ground-truth' methane
- L302: ... was expected 'to' emit ...
- L306: please delete 'and'
- L316: 400 ppb is roughly the range of plume enhancements shown in Figure 6, but not Figure 9.

- L325-326: How can you assess a correlation between C2H6 and (delete '&') CH4 when C2H6 enhancements cannot be discerned from background levels due to a low signal to noise ratio?
- L365: "with the error associated for other low-emission" needs to be revised for clarity

---

## Referee Comment (RC2)

Review: Dooley et al. AMT 2024.

General Comments:

There is indeed a gap that requires filling for methodologies to accurately quantify methane at point and site level for sub 50kg/hr emission rates – which this type of method can help to address. There is plenty of good work in the manuscript and really just needs some refinement and expansion to be suitable for full publication.

Main specific comments:

Wind:

There is a lack of discussion around the accuracy of the on-board wind measurements. As the wind measurement is so critical to the mass balance methodology and there is other work discussing the challenges of accurate wind measurements from drones, I would have expected a more comprehensive discussion on the particular nature of the set up and the uncertainties associated with different wind speeds and movement speed of the drone.

Have there been direct comparisons to mast or tower measurements of wind speeds and directions? Are there periods in flight where the winds are clearly no longer correct due to either drone motion or wind speeds? Should there be filtering applied to certain events (e.g fast turns?). These bits of specific information are important as other groups may use papers such as this as templates for setting up their own systems, or alternatively commercial outfits may be referring back to work such as this for justifying uncertainties when performing legally complying work for future methane regulations.

Introduction:

The introduction feels like it is somewhat out of date, there is a lack of recent references compared to the rest of the manuscript. Given the nature of the package being demonstrated here, I believe that there should also be reference to the platforms and packages that have been developed in the commercial sector as well as academia.

Abstract:

There is a bit too much general background in the abstract for my liking on the global importance of $CH_4$ which really just belongs in the introduction. I'd prefer to see the abstract with some extra important technical details on the package, such as flight time capability and the limitations of the flying conditions in which good results were achieved.

Methods / Uncertainties:

I have concerns that it is not completely clear to me how the extrapolation to the top and bottom of the plume are computed (e.g. Figure 9) when it is clear that none of the transects are in background air at the lowest and highest transect. Similarly, it is not clear that the plume edge is caught on the lower most transects in plume 9. How are these issues accounted for, and how are you able to ascribe uncertainty to that unknown?

Would it be possible to have an SI with figures to show the transects / plume cross-sections for all the calculated emission rates to inform the reader visually with the data quality?

Minor comments:

L40-55: More discussion on wind parameters and uncertainty within the context of mass-balance feels needed.

L60: Satellite systems feels far too catch all. Please separate out into point source and area mapper discussion and allocate ranges of capability accordingly (it doesn't feel right to lump GHGsat with TROPOMI in this type of discussion).

L84: Precision discrepancy with abstract and later in manuscript. Please check and be clear where numbers are quoted from field measurements and where just taking manufacturers stated values.

L86: Define response time – is this a 1/e value, a 90% fall time, some other metric?

L178: What criteria was used to determine the order of the polynomial fit? Are there issues with fitting to the beginning and end of the run using higher order polynomials?

L186 and Fig 4: Different references are given to the stability class references – please ensure that these are correct or if both should be referenced at both locations. More information on what is going into the stability class selection would be helpful (maybe SI worthy rather than main script?)

L265: Define standard flight conditions!

Fig 8: It would feel better to quote the uncertainties (and maybe throughout) to 2 sigma – it would make the figure more compelling that the method is reliable to 95% confidence.

Fig 11 and Ethane:methane discussion. (Calibration?)

The data within this figure looks fantastic, but belies an issue that at no point has calibration of the $CH_4$ or $C_2H_6$ been discussed. From personal experience, I have seen similar sensors have gain factors of 0.7 on one channel (which would then make a significant difference to the results) – so I would hope that the instrument has been functionally calibrated in the laboratory prior to deployment. It would be expected that the calibration routine is at least alluded to in the manuscript, and potentially the details of the calibration put in the SI. I am mainly asking for this so that other groups do not use these instruments expecting that absolutely zero calibration is required.

---

## Author Comment (AC1)

**2 Response to Reviewer Comments 2 (RC2)**

Link to Original Submission: A New Technique for Airborne Measurements to Quantify Methane Emissions Over a Wide Range: Implementation and Validation – Dooley et al. (2024)

**(RC 201)** There is a lack of discussion around the accuracy of the on-board wind measurements. As the wind measurement is so critical to the mass balance methodology and there is other work discussing the challenges of accurate wind measurements from drones, I would have expected a more comprehensive discussion on the particular nature of the setup and the uncertainties associated with different wind speeds and movement speed of the drone.

**(AC 201)** *L186-199. Section 2.2. Figure A1. Figure A2.* We thank the reviewer for identifying this weakness in the methodology presentation. The accurate measurement of winds can be challenging, and we have added content to the paper that better describes how the flight data are screened to remove wind anomalies arising from vertical motions or rapid horizontal accelerations of the M600P platform. We have added two figures to a supplement section (figures A1 and A2), and section 2.2 in Methods now includes a more detailed discussion of static wind calculation and validation.

**(RC 202)** Have there been direct comparisons to mast or tower measurements of wind speeds and directions? Are there periods in flight where the winds are clearly no longer correct due to either drone motion or wind speeds? Should there be filtering applied to certain events (e.g., fast turns?). These bits of specific information are important as other groups may use papers such as this as templates for setting up their own systems, or alternatively commercial outfits may be referring back to work such as this for justifying uncertainties when performing legally complying work for future methane regulations.

**(AC 202)** *L186-199.* See final paragraph in section 2.2.

**(RC 203)** The introduction feels like it is somewhat out of date, there is a lack of recent references compared to the rest of the manuscript. Given the nature of the package being demonstrated here, I believe that there should also be reference to the platforms and packages that have been developed in the commercial sector as well as academia.

**(AC 203)** *L64-94.* An overview of other UAS methods has been added to section 1.

**(RC 204)** There is a bit too much general background in the abstract for my liking on the global importance of $CH_4$ which really just belongs in the introduction. I'd prefer to see the abstract with some extra important technical details on the package, such as flight time capability and the limitations of the flying conditions in which good results were achieved.

**(AC 204)** *L16-31.* First two paragraphs of section 1 were combined and merged into L16-31.

**(RC 205)** I have concerns that it is not completely clear to me how the extrapolation to the top and bottom of the plume are computed (e.g., Figure 9) when it is clear that none of the transects are in background air at the lowest and highest transect. Similarly, it is not clear that the plume edge is caught on the lowermost transects in Figure 9. How are these issues accounted for, and how are you able to ascribe uncertainty to that unknown?

**(AC 205)** *L321-334. Section 2.6.* The reviewer has identified a critical aspect which should be made more clear – that incomplete plume sampling is one of the major challenges of implementing this technique. Please see more detailed discussion in section 2.6.

**(RC 206)** L60: "Satellite systems" feels far too catch-all. Please separate out into point source and area mapper discussion and allocate ranges of capability accordingly (it doesn't feel right to lump GHGsat with TROPOMI in this type of discussion).

**(AC 206)** *Section 1. L44-50. Section 4. Figure 12.*

**(RC 207)** L84: Precision discrepancy with abstract and later in manuscript. Please check and be clear where numbers are quoted from field measurements and where just taking manufacturers stated values.

**(AC 207)** *Table 1.* This error is corrected in the updated manuscript, uncertainty values quotes throughout the document now match the uncertainties listed table 1.

**(RC 208)** L86: Define response time – is this a $1/e$ value, a 90% fall time, some other metric?

**(AC 208)** *L112-115.* The "response time" was referring to the time needed for samples to travel from inlet to MIRA. Section 2.1.4 has been rewritten for clarity.

**(RC 209)** L178: What criteria was used to determine the order of the polynomial fit? Are there issues with fitting to the beginning and end of the run using higher order polynomials?

**(AC 209)** *L229-231.* The polynomial order, and other parameters described in section 2.3 selected empirically (trial-and-error) during initial processing. Higher order polynomials can affect the edges of the fitted timeseries; generally, however, higher order polynomials are not required after appropriately setting the other threshold parameters used for the gradient and outlier filters.

**(RC 210)** L186 and Fig 5: Different references are given to the stability class references – please ensure that these are correct or if both should be referenced at both locations.

**(AC 210)** *L251-254. Figure 6.* The Gaussian model used in section 2.4 was taken from Seinfeld et al. 2006. The stability class parameters are available in Seinfeld et al. 2006 and further detailed in Woodward 2010. Both references should be cited whenever stability classes are discussed in the text.

**(RC 211)** More information on what is going into the stability class selection would be helpful (maybe SI worthy rather than main script?).

**(AC 211)** *Section 2.4.* The Gaussian plume models used in section 2.4 helped with flight preparation and visualization, but detailed discussion of modeling parameters is outside the scope of this work.

**(RC 212)** L265: Define standard flight conditions!

**(AC 212)** *L366-369.* We define standard flight conditions as periods of high solar insolation and steady windspeed between 2-6 m/s. This is discussed on lines L366-369 in the context of calculating the Limit of Detection (LOD) for the system. The plume evolution is dependent on multiple environmental conditions, but a cross-sectional area of 100 m2 is used in the LOD calculation.

**(RC 213)** Fig 8: It would feel better to quote the uncertainties (and maybe throughout) to $2\sigma$ – it would make the figure more compelling that the method is reliable to 95% confidence.

**(AC 213)** *Throughout. Figure 8. Figure 12. Table B1.* The $2\sigma$ (95%) confidence interval is used for all presented flux uncertainties and figures.

**(RC 214)** Fig 11 and Ethane:methane discussion. (Calibration?) The data within this figure looks fantastic, but belies an issue that at no point has calibration of the $CH_4$ or $C_2H_6$ been discussed. From personal experience, I have seen similar sensors have gain factors of 0.7 on one channel (which would then make a significant difference to the results) – so I would hope that the instrument has been functionally calibrated in the laboratory prior to deployment. It would be expected that the calibration routine is at least alluded to in the manuscript, and potentially the details of the calibration put in the SI. I am mainly asking for this so that other groups do not use these instruments expecting that absolutely zero calibration is required.

**(AC 214)** *L265-272.* A controlled release 'pulse' from a known natural gas source is used to measure any lag or gain offset between the $CH_4$ and $C_2H_6$ channels.

---

## Author Comment (AC2)

**1 Response to Reviewer Comments 1 (RC1)**

Link to Original Submission: A New Technique for Airborne Measurements to Quantify Methane Emissions Over a Wide Range: Implementation and Validation – Dooley et al. (2024)

**(RC 101)** The title is a bit vague in referencing a "wide range" – it is referenced that the UAS can sample scales of up to 1 km, but aircraft can sample at much wider ranges... would "A new UAS-based technique for quantifying and attributing methane emissions from small and distributed point sources: [...]" or similar be more relevant?

**(AC 101)** *Title.* We appreciate the reviewer's suggestion on the title and have updated it to "A New Aerial Approach for Quantifying and Attributing Methane Emissions: Implementation and Validation". This removes the ambiguous "wide range" statement and better summarizes the system's goals and development.

**(RC 102)** In general, the manuscript is organized appropriately, but structuring within sections could be either reconfigured, renamed, or provide more detail. In general, it could be helpful to add a traditional "Methods" section, which encompasses Sections 2-3, whereas "Results" can include section 4.

**(AC 102)** *Section 2. Section 3.* The original sections 2 (System Design) and 3 (Deployment) were combined into a single "Methods" Section (2). Original section 4 was renamed from "Analysis" to "Results" (Section 3).

**(RC 103)** The information in L110-116 seems more like "onboard data logging and transmission".

**(AC 103)** *Section 2.1.5. L162-168.* This section was renamed as proposed and moved to the "Methods" (Section 2).

**(RC 104)** Section 2.5: please explain why this section is needed; L193-195 provides the motivation for this paragraph, but this motivation (and perhaps L196-203) should be described and presented at the beginning of this section.

**(AC 104)** *L241-247.* A paragraph motivating the plume simulations was added to the beginning of Section 2.4.

**(RC 105)** Section 4: this section is difficult to follow with the many flights that occurred at different times, with varying purposes (e.g., October/November 2022 controlled release experiments; Socorro, NM MWF, Spring 2022-Summer 2023; Orphan Well, April 2023; WWTP Summer 2023). A table outlining the flights, locations, or purpose, or even more descriptive section headers and a few sentences describing each might help. To me, the distinction between (a) controlled-release experiments and validation of the system and (b) smaller, point source emissions detection via case studies is important.

**(AC 105)** *Section 3.1. Section 3.2. Table B1.* This section was split into clearer sections and subsections to highlight the (a) controlled release validation flights (section 3.1) and (b) the smaller targeted source case studies (section 3.2). Additionally, a table with information on source type, environmental conditions, and estimated emission rates was added to the manuscript (table B1)

**(RC 106)** Background determination: L165-169 could be expanded upon to detail the procedure that is plotted in Figure 3. Please also describe the gradient method you use in (3b) within the text (and not just the caption) and how this is used in (3c).

**(AC 106)** *L218-234.* These paragraphs were rewritten to better describe the background estimation process and how each of the steps follows from the previous.

**(RC 107)** Figure 7: This can be described more effectively, in general. If I am reading this correctly, this is the uncertainty in the background estimation from just two test flights, which ties to Section 2.4 and is better explained prior to the uncertainty estimate in Section 3.3. To me, this figure and its description would be better suited just following Figure 3, where readers can directly connect the uncertainty in background $CH_4$ to $CH_4$ emissions rate uncertainties.

**(AC 107)** *Figure 4.* This figure was moved to section 2.3, as suggested, since the error confidence intervals shown directly follow from the background estimation routine. Figure 4 was updated to include three different flights in order to highlight the variability in background estimation residuals. Additionally, the average confidence interval from 28 independent flights is now overlaid on this plot for reference.

**(RC 108)** Do you have estimates of how results in Figure 7 compare to all flights (i.e., are the two test flights representative of typical flights)? Why is the background uncertainty not incorporated on a per-flight basis?

**(AC 108)** *Figure 4. Table 1. L235-238.* The original plot was updated to include a third representative flight as well as the average background estimation confidence interval from 28 independent flights. Comparison with the average confidence interval shows that these individual flight residuals, while different, are representative of the mean. The reviewer is correct that all background uncertainties are applied to the flux quantification error analyses on a per-flight basis, Thus, the average uncertainties due to background estimations in Table 1 are provided only as representative values for these quantities. The manuscript has been revised to better emphasize these points.

**(RC 109)** The word "baseline" is used interchangeably with "background" and it would be clearer in the text to just use one or the other.

**(AC 109)** *Throughout.* For consistency, the word "background" is now used throughout the manuscript as opposed to "baseline".

**(RC 110)** Section 3.3: this section needs to be expanded upon so that it is clear where uncertainties in each term in Equations 4-6 come from and how they contribute to the overall uncertainty in $F_{tot}$. For example, Table 1 seems to only describe onboard UAS sensor precision and/or accuracy, but not include other sources of uncertainty like the uncertainty in $CH_4$ and $C_2H_6$ enhancements, or uncertainties in $(\boldsymbol{u} \cdot \hat{n})$. All of these propagated uncertainties should be incorporated in the lower LOD of the flux estimate, correct?

**(AC 110)** *Section 2.7. L342-365. Section B1.* The reviewer is correct that all the propagated uncertainties are incorporated into the final flux uncertainty and lower limit of detection. The paper was unclear on this point and more complete details are now given in. section 2.7 of the updated manuscript.

**(RC 111)** L96: wind speed magnitude uncertainty is $0.35\,\mathrm{m/s}$, but Table 1 states $0.2\,\mathrm{m/s}$. Please describe how the wind speed uncertainty is derived.

**(AC 111)** *L120. Section 2.7. Table 1. Section B1.* The individual vector wind speed uncertainty is $0.2\,\mathrm{m/s}$, and the text (L120) and Table 1 are now consistent.

**(RC 112)** The MIRA $CH_4$ uncertainty is stated as 10 ppb here, but the error in background derivation is stated as 20 ppb. Is the propagation of MIRA $CH_4$ precision and the background uncertainty taken into account in $F_{tot}$, which likely adds to the overall uncertainty and lower LOD? It is unclear how the background uncertainty from each flight is incorporated into the total flux uncertainty. All of this would be very advantageous to outline in Section 3.3, and similarly, for $C_2H_6$.

**(AC 112)** *Section 2.7. Table 1.* The uncertainties in table 1 have been updated for clarity and the per-flight error propagation is further detailed in Section 2.7.

**(RC 113)** It should be stated somewhere how the MIRA is calibrated prior to each flight for $CH_4$ and $C_2H_6$.

**(AC 113)** *L265-272.* The response of the MIRA Pico is evaluated prior to each flight by releasing a small volume of processed natural gas approximately $1\,\mathrm{m}$ upwind of the gas inlet tube. We verify that pulses are recorded in both methane and ethane, and check for time coincidence and consistency in ethane/methane ratio. This procedure is now described in the paper.

**(RC 114)** Please state what is meant by "standard flight conditions".

**(AC 114)** *L366-369.* We define standard flight conditions as periods of high solar insolation and steady windspeed between 2-6 $\mathrm{m/s}$. This is discussed on lines L366-369 in the context of calculating the Limit of Detection (LOD) for the system. The plume evolution is dependent on multiple environmental conditions, but a cross-sectional area of 100 m2is used in the LOD calculation.

**(RC 115)** The uncertainties for each of the various source emission rates calculated in section 4 should be stated alongside the measured emissions rates. Some are stated, but others are not (e.g., in 4.1.2). This is important when assessing how well the technique might fare with one type of source versus another.

**(AC 115)** *Throughout. Table B1.*

**(RC 116)** Why is the upper uncertainty in 4.1.3 for the WWTP up to 250%, and why does the lower uncertainty differ? Because this is an AMT manuscript, it would benefit the reader to offer explanations for the calculated uncertainties in each of the various cases to assess limitations and capabilities of this technique.

**(AC 116)** *Figure 12. Table B1.*

**(RC 117)** L289: The results of the controlled release experiments indicate that there is a "systematic" underestimation, but this does not look systematic as some estimates are higher than the metered emissions rates. Can the setup of the experiment be explained in more detail here? The type of point source is mentioned, with wind conditions, but how was the UAS flown and does this contribute to this underestimation? There are no hypotheses provided for why emissions estimates from the UAS system are lower in general – can you provide some?

**(AC 117)** *Section 3.1. L393-396. L321-334.* Section 3.1 has been updated and possible causes of the underestimation are discussed at L393-396 and L321-334.

**(RC 118)** "UAS" is used, but also "UAV" – please choose one or the other to be consistent.

**(AC 118)** *Throughout.* Uncrewed Aerial Vehicle (UAV) is the mobile platform, the Matrice 600 Pro (M600P) alone. Uncrewed Aerial System (UAS) is the complete instrument including mounted payload and data acquisition hardware/software. The authors recognize that this is a subtle difference and have therefore opted to use "M600P" and "UAS" to refer to the vehicle and the complete system, respectively.

**(RC 119)** Figure 1, caption, and L126: What does "dual-opening" mean with respect to the sampler inlet?

**(AC 119)** *L149-150. Figure 1.* "Dual-opening" refers to the y-shaped inlet on the sampler port which decreases the chance of damage to the MIRA pump due to clogging from dust and debris. This wording has been changed in 1 and better described in L149-150.

**(RC 120)** L84: precision on $CH_4$ and $C_2H_6$ is not what is specified in the abstract – please confirm which is correct.

**(AC 120)** *Abstract. Table 1.* This was an error and has been updated in the updated report.

**(RC 121)** L86: below, the response time was ~2 s – please state one or the other for consistency.

**(AC 121)** *L112.* This was an error due to changes in length of the sampler tubing and mast inlet position. The correct phase delay between inlet and MIRA is approximately 2 seconds due to the pump flow rate and tube length.

**(RC 122)** L90: mole fractions are presented here in ppm or ppb, which is a mol/mol. Please correct either mixing ratio or mole fraction determinations throughout.

**(AC 122)** *Throughout.* Mole fraction and mixing ratio were used inappropriately and interchangeably in the original submission. Units of ppm and ppb are $mol_{compound}/mol_{air}$ and the more correct term "mole fraction" is used throughout the updated manuscript.

**(RC 123)** L141: symbol for 'yaw' different here than it is in equation 1.

**(AC 123)** *L170.* Typographic error in section 2.2 has been corrected in the updated manuscript.

**(RC 124)** Figure 12: What do the individual colors mean? If nothing, does it make sense to have controlled release vs. municipal vs. orphan vs. WWTP all be different colors?

**(AC 124)** *Figure 8. Figure 12.* The individual colors were unimportant and misleading. Thank you for the recommendation and figures 8 and 12 have been updated.

**(RC 125)** Figure 6: $\chi - \chi_0$ is used to show an enhancement, whereas in L338, a delta symbol is used, in addition to both being used in Figure 10 – please choose one or the other for clarity.

**(AC 125)** *Throughout.* $(\chi - \chi_0)$ is now used throughout all images and text in the updated manuscript.

**(RC 126)** L325-326: How can you assess a correlation between $C_2H_6$ and (delete '&') $CH_4$ when $C_2H_6$ enhancements cannot be discerned from background levels due to a low signal-to-noise ratio?

**(AC 126)** *Figure 10. L429-442.* The reviewer brings up a very good point which was not discussed in the initial submission. Figure 9 has been updated to include the expectation (modeled) $C_2H_6$ timeseries corresponding to the measured $CH_4$ for two different thermogenic mixtures.

**(RC 127)** L365: "with the error associated for other low-emission" needs to be revised for clarity.

**(AC 127)** *Section 4. L491-493.*

---

## Author Comment (AC3)

**3 Response to Community Comments 1 (CC1)**

Link to Original Submission: A New Technique for Airborne Measurements to Quantify Methane Emissions Over a Wide Range: Implementation and Validation – Dooley et al. (2024)

**(RC 301)** Line 150: Authors are neglecting the effect of the pitch and roll angles on the wind measurement of Trisonica mini which is not a bad estimation if the copter does not pitch and roll during flight (i.e., maybe only during hovering at a certain altitude). However, in line 211, the flight speed was given as between 2-5 m/s which I think will force the copter to roll or pitch at about 10-15 degrees.

**(AC 301)** *L186-199. Figure A1.* The average pitch and roll of the UAS is on the order of 1-3 degrees during steady, level flights through the plume (see Figure A1). The pitch and roll increases when the UAS accelerates (e.g., at the ends of each transect) but these samples are filtered out during processing, prior to flux estimation.

**(RC 302)** Considering the placement of the anemometer (~0.8 m over the propeller plane), the angular momentum might become non-negligible I think. Therefore, there might be a bias in the wind speed measurements during the flight. I think this should be clarified, if there is a small effect this needs to be shown by the authors. I think checking Donnel et al. (2018) paper might help with this (https://doi.org/10.2514/6.2018-2986).

**(AC 302)** *L186-199.* Samples collected during maneuvers causing large pitch and roll are filtered out during processing, prior to any flux calculations. The measured pitch and roll for the entirety of each flight shown in Figure 12 and Table B1 are plotted in figure A1.

**(RC 303)** Figure 3, from the figure it looks like the widths of each spike are not similar. If the flight characteristics are the same for each repeated crosswind flight why then the width of these spikes are different? Is this because of the environmental conditions?

**(AC 303)** *L215-217.* Each 'spike' in figure 3 is a plume concentration measurement from an individual transect. Variability in spike magnitude and duration is due to different transect altitude and downwind locations relative to the target source.

**(RC 304)** Additionally, why does the background $CH_4$ increase over time? $CH_4$ concentrations are between 2.2-2.3 ppm before 800 s, and it increases at about 2.5 ppm at the end of the measurement. I would expect that the background signal will be more or less similar during the flight and when the drone sees the plume the spikes will occur. Maybe this figure (3) needs more explanation.

**(AC 304)** *Figure 7. L213-238.* The steady increase in $CH_4$ was due to changes in local environmental conditions during collection.

**(RC 305)** Line 259: Why was the flight conducted 130 m away from the source? Why not closer, were there any restrictions?

**(AC 305)** *Section 2.5. Section 4. L480-490.* Distance is a function of local topography and environmental conditions, in the case of figure 7 there was a 60 m perimeter around the source (controlled release stack) and a further 60-70 m south to allow more growth.

**(RC 306)** Line 265: This is a bit cryptic. How did the authors find the lower quantification threshold here? What are the standard flight conditions?

**(AC 306)**  *L366-369.*  We define standard flight conditions as periods of high solar insolation and steady windspeed between 2-6 m/s. This is discussed on lines L366-369 in the context of calculating the Limit of Detection (LOD) for the system. The plume evolution is dependent on multiple environmental conditions, but a cross-sectional area of 100 m2is used in the LOD calculation.

**(RC 307)**  Also, in Table 1, how did the authors come up with the Mira Pico uncertainties? When I check the manual, the only information given about the instrument is the sensitivity which is $< 1\,\mathrm{ppb/s}$ and the drift which is given as $30\,\mathrm{ppb}$. Maybe adding a bit more explanation for Figure 7 might help here.

**(AC 307)**  *Section 2.7. Table 1. L342-350.*  The MIRA Pico used in this study (circa 2019) is stated as having sensitivity of $1\,\mathrm{ppb\,s^{-1}}$ $CH_4$ and $0.5\,\mathrm{ppb\,s^{-1}}C_2H_6$. This is an excellent sensitivity level on the raw concentration measurements $(\chi)$ and the quasi-periodic drift is removed using the background estimation and removal $(\chi_0)$ as described in section 2.3. The later processes introduce a $3\sigma$ error of ~16 ppb $CH_4$, ~2.5 ppb $C_2H_6$.